# From Pixels to Factors:
# Learning Independently Controllable State Variables for Reinforcement Learning

## Abstract

Algorithms that exploit *factored* Markov decision processes are far more sample-efficient than factor-agnostic methods, yet they assume a factored representation is known *a priori*—a requirement that breaks down when the agent sees only high-dimensional observations. Conversely, deep reinforcement learning handles such inputs but cannot benefit from factored structure. We address this representation problem with Action-Controllable Factorization (ACF), a contrastive learning approach that uncovers *independently controllable* latent variables—state components each action can influence separately. ACF leverages sparsity: actions typically affect only a subset of variables, while the rest evolve under the environment's dynamics, yielding informative data for contrastive training. ACF recovers the ground-truth controllable factors directly from pixel observations on three benchmarks with known factored structure—Taxi, FourRooms, and MiniGrid-DoorKey—consistently outperforming baseline disentanglement algorithms.

## 1 Introduction

Over the last decade, deep reinforcement learning (RL) has enabled agents to learn complex behaviors directly from high-dimensional observations—e.g., pixels in Atari games (Mnih et al., 2015), and continuous control from pixels (Lillicrap et al., 2016; Levine et al., 2016)–without manual feature engineering. However, this flexibility comes at a cost: modern deep RL methods remain strikingly sample-inefficient.

Classical work in factored RL shows that, if the underlying Markov decision process (MDP) can be decomposed into state-variable factors with sparse dependencies, one can achieve exponential gains in both model learning and planning (Boutilier et al., 1995; Guestrin et al., 2003). Indeed, factored variants of PAC-RL algorithms such as factored $E^3$ (Kearns & Koller, 1999) and Factored RMax (Guestrin et al., 2002; Brafman & Tennenholtz, 2002), provably exploit these structures for faster convergence, and subsequent methods even learn the dependency graph online (Strehl et al., 2007; Diuk et al., 2009). More recently, factored representations have proven useful for world modeling (Wang et al., 2022b; Pitis et al., 2020; 2022), exploration (Wang et al., 2023; Seitzer et al., 2021), and skill discovery (Vigorito & Barto, 2010; Wang et al., 2024; Chuck et al., 2024; 2025). Crucially, all these gains depend on having access to a hand-specified factored representation.

State-of-the-art model-based deep RL approaches avoid simulating trajectories in raw observations by learning latent world models end-to-end (Hafner et al., 2019; Schrittwieser et al., 2020; Hansen et al., 2022; Rodriguez-Sanchez & Konidaris, 2024). Some efforts attempt to induce factorization in these latent spaces (Hansen et al., 2022; Hafner et al., 2025), but they do not offer empirical or theoretical guarantee that any factor is identified. In parallel, unsupervised and self-supervised learning has long studied disentanglement (Bengio et al., 2013; Locatello et al., 2019) as a way to achieve better generalization. Although there is no consensus formalization of disentanglement, two classical approaches are nonlinear ICA (Independent Component Analysis; Comon (1994); Hyvärinen et al. (2023)) and causal representation learning (Schölkopf et al., 2021) yet these methods do not fully ground to decision making and control. For instance, some methods pretrain disentangled representations for RL (Higgins et al., 2017a) based on the assumption that the learned variables will

be useful for downstream decision-making. Meanwhile, causal representation learning (Schölkopf et al., 2021) leverages the notion of interventions, related to an RL agent actions, but do not directly address the sequential decision making problem.

We address this representation gap for factored RL by explicitly targeting the recovery of independently controllable variables (Thomas et al., 2018). Our key idea is to use a contrastive objective that compares the predicted next-state distributions under agent actions against those under the environment's natural dynamics. Thus, we align our latent factors with the underlying state variables that are controlled independently. We validate our approach on pixel-based versions of Taxi (Dietterich, 2000), MiniGrid-DoorKey (Chevalier-Boisvert et al., 2023; Pignatelli et al., 2024), and FourRooms (Sutton et al., 1999), showing that we can automatically recover controllable state variables that align with an expert-designed representation, directly from the pixels.

**Contributions** First, we formalize the representation learning problem for factored RL as the problem of identifying *independently controllable* latent variables. Moreover, we propose a novel contrastive learning objective that leverages action-induced discrepancies in next-state predictions to isolate controllable factors. Finally, we demonstrate empirically that our method recovers ground-truth controllable factors directly from pixels in classical RL domains.

## 2 BACKGROUND

**Markov Decision Process** We consider an agent that acts in a Markov Decision Process (MDP; Puterman (1994)) $\mathcal{M} = \langle \mathcal{S}, A, T, R, p_0, \gamma \rangle$ with a continuous state space $\mathcal{S} \subseteq \mathbb{R}^{d_s}$ and a discrete action set $A$. The transition function $T : \mathcal{S} \times A \to \Delta(\mathcal{S})$[1] models the world's dynamics, $R : \mathcal{S} \times A \to \mathbb{R}$ is a reward function, $\gamma \in [0, 1)$ is the discount factor, and $p_0 \in \Delta(\mathcal{S})$ is the initial state distribution.

**Factored MDPs** (FMDPs; Boutilier & Dearden (1996)) are a particular class of MDPs that have a factorized transition function:

$$T(s' \mid s, a) = \prod_{i=1}^{K} T_i(s_i' \mid \mathrm{pa}(s_i'), a),$$

where the state space is $\mathcal{S} = \mathcal{S}_1 \times \cdots \times \mathcal{S}_K$ and each $S_i \subseteq \mathbb{R}$. Moreover, $\mathrm{pa}(s_i') : \mathcal{S}_i \to \mathcal{P}([K])$, where $\mathcal{P}([K])$ is the power set of $[1, 2, \ldots, K]$, and it represents the set of factors required to predict $s_i'$. Typically, this structure is represented by a Dynamic Bayesian Network (DBN; Boutilier et al. (1995; 2000)).

**Nonlinear Independent Component Analysis** (ICA; Comon (1994)) is the problem of identifying a set of independent signal sources from entangled measurements. Formally, given a set of generating sources $\{s_i \in \mathcal{S}_i\}_{i=1}^{K}$ that are independent and distributed according to densities $p(s_i)$, ICA identifies the set of generating signals from observed measurements $x$ that are entangled (mixed) by an unknown function $o$—i.e., $x = o(s_1, \ldots, s_K)$. In the case of non-linear ICA, $o$ is a nonlinear, invertible function. In particular, we will consider the problem of non-linear ICA with auxiliary variables (Hyvärinen et al., 2019), where the sources to be identified are *conditionally* independent given an auxiliary variable $u$. Moreover, we assume that our agent receives high-dimensional observations that are generated by an observation function that is a diffeomorphism[2] $o : \mathcal{S} \to X \subseteq \mathbb{R}^{d_x}$, where $d_x \gg d_s$.

## 3 ACF: ACTION CONTROLLABLE FACTORIZATION

Imagine a simple desk lamp with two separate switches: one toggles the lamp's power, flipping it on or off, while the other cycles the bulb's color between warm and cold light. If you leave both switches untouched, the lamp may still occasionally flicker on or change color on its own, but with a significantly lower probability. By observing the lamp when you flip only the power switch versus doing nothing, you isolate the "on/off" factor; likewise, by pressing only the color switch versus leaving it alone, you isolate the "color" factor. Because each switch only affects one property while the other property evolves naturally, you can disentangle these two characteristics simply by

---

[1] $\Delta(X)$ is the set of probability densities over set $X$.

[2] A bijection that is continuously differentiable and whose inverse is also continuously differentiable.

contrasting action-driven changes with natural behavior. The lamp might have other characteristics like volume, weight, and shape; however, these are not factors that can be controlled by the agent. Here we focus on disentangling factors that *are* controllable.

### 3.1 PROBLEM FORMULATION

**Setting** We consider that the agent does not have access to the ground truth factored state space $S$. Instead, it gets high-dimensional observations that are generated by an unknown diffeomorphism $o : \mathcal{S} \to X \subseteq \mathbb{R}^{d_x}$. Hence, we are concerned with learning from the observed samples of $T(x' \mid x, a)$ an encoder $f_\phi : X \to Z$, where $Z$ factorizes as $Z = Z_1 \times \cdots \times Z_K$, that *identifies* the underlying factors.

**Identification** Formally, we say that a learned factorization identifies the underlying factor $\mathcal{S}_i$ if and only if there exist invertible functions $h_i$ and permutation function $\rho$ such that $h_i : Z_i \to S_{\rho(i)}$ for all $i \in \{1, \ldots, K\}$. That is, we can recover the underlying factors up to permutation and invertible transformations.

In many problems, the agent's actions have sparse effects on the environment: just a few factors are controlled, while others just follow their natural transition, unaffected by the agent. To help the agent understand its environment, we assume that the agent has a *special action* $a_0$ that corresponds to a *no-op* (or observe) action that allows the agent to observe the natural evolution of the environment without intervening.

**Transition Dynamics** Let $\Psi : \mathcal{S} \times A \to \mathcal{P}([1, 2, \ldots, K])$ be the set of variables affected by action $a$ in state $s$. We assume the transition dynamics factorize as follows,

$$T(s' \mid s, a) = \prod_{i \in \Psi(s,a)} T(s_i' \mid s, a) \prod_{j \notin \Psi(s,a)} T(s_j' \mid s, a_0); \tag{1}$$

where $T(s_i' \mid s, a_0)$ represents the natural (or observational) dynamics. In here, we will consider conditioning the transition dynamics on the full current state $s$, instead of just the parents, given that $T(s_i' \mid s, a) = T(s_i' \mid \mathrm{pa}(s_i'), a)$.

Moreover, for the unknown observation function $o$, a diffeomorphism, we know that the *observed* dynamics follow (Boothby, 2003):

$$T(x' \mid x, a) = |\det \left( J_{o^{-1}}(x')^T J_{o^{-1}}(x') \right)|^{1/2} T(s' \mid s, a), \tag{2}$$

This equation relates the observed dynamics $T(x' \mid x, a)$ to the underlying ground truth state dynamics $T(s' \mid s, a)$ by the Jacobian matrix $J_{o^{-1}}$, whose determinant quantifies the change in volume between the two spaces. This relation can be seen as the generalization to higher dimensions of the change of variable formula in probability theory.

### 3.2 ALGORITHM

**Energy Parameterization** We parameterize the encoder by $f_\phi(x) \mapsto z$, with parameters $\phi$, and, more importantly, we parameterize the transition function as the sum of energy functions (unnormalized probability densities) such that, $T(z' \mid z, a) \propto \exp \left( \sum_{i=1}^{K} E_\theta(z_i', a, z) \right)$ with $i \in [K]$ and parameters $\theta$. This sum of energies reflects the factorized structure where each energy represent the transition dynamics of latent variable $z_i$.

**Learning a Markov Representation** In order to estimate these energy functions from data and learn a Markov representation suitable for RL (Allen et al., 2021), we optimize the following training objectives. Firstly, we estimate the inverse dynamics $I^\pi$ using our energy functions, as follows,

$$I^\pi(a \mid z, z') = \frac{T(z' \mid z, a)\pi(a \mid z)}{\sum_{a'} T(z' \mid z, a')\pi(a' \mid z)} \propto \frac{\exp \left( \sum_i E_\theta(z_i', a, z) \right) \pi(a \mid z)}{\sum_{a' \in A} \exp \left( \sum_i E_\theta(z_i', a', z) \right) \pi(a' \mid z)}; \tag{3}$$

and because our action set is discrete, we can use a softmax multiclass classifier to learn our inverse function by minimizing the cross entropy loss:

$$\mathcal{L}_{\mathrm{inv}}(\phi, \theta) = -\log I^\pi(a \mid z, z'). \tag{4}$$

Secondly, we use InfoNCE (Oord et al., 2018) to maximize the mutual information between $z$ and $z'$: we use a batch $B$ of $N-1$ negative samples and 1 positive sample, and minimize the following loss,

$$\mathcal{L}_{\text{fwd}}(\phi, \theta) = -\log \frac{\exp\left(\sum_i E_\theta(z'_i, a, z)\right)}{\sum_{z^j \in B} \exp\left(\sum_i E_\theta(z_i^j, a, z)\right)}. \tag{5}$$

Optimizing these losses guarantee that we learn a Markov representation that preserves the relevant information for action effects prediction (Allen et al., 2021) without requiring an explicit reconstruction loss. However, they do not ensure that the representation will align with the controllable factors.

To see this, consider an invertible mapping $g : S \to Z$ between the ground truth state $s$ and another representation $z$. The relation between the densities is given by the following change of variable formula: $T(s' \mid s, a) = |\det J_g(s')| T(z' \mid z, a)$. Therefore, if $|\det J_g(s')| = 1$ (e.g., $g$ is a rotation), the distribution will match even in the case we use a factorized prior (for an extended discussion, see Locatello et al. (2019); Hyvärinen et al. (2023))

**Factorizing the Controllable Variables** We formalize our intuition and exploit the sparsity of the actions' effects to learn a latent representation $Z$ that identifies the controllable factors.

The core idea is to contrast the effect of an action, the distribution $T(x' \mid x, a)$, against the natural dynamics $T(x' \mid x, a_0)$, where $a_0$ is the no-op action, using the following ratio:

$$\log r_a(x', x) = \log \frac{T(x' \mid x, a)}{T(x' \mid x, a_0)} = \log \frac{|\det(J_{o^{-1}}(x')^T J_{o^{-1}}(x'))|^{1/2} \prod_i T(s'_i \mid s, a)}{|\det(J_{o^{-1}}(x')^T J_{o^{-1}}(x'))|^{1/2} \prod_i T(s'_i \mid s, a_0)};$$

$$= \log \frac{T(s'_j \mid s, a)}{T(s'_j \mid s, a_0)} = \log r_a(s', s),$$

where $s'_j$ is the factor affected by $a$ when executed in $s$. Therefore, this ratio is a function of the factor $s'_j$ and not the rest.

In practice, we can estimate these ratios from observed transitions contrastively (Gutmann & Hyvärinen, 2010; Hyvärinen et al., 2019). We leverage our energy parameterization to infer a binary classifier that differentiate between transitions from action $a$ from another, e.g. the null action $a_0$.

These classifiers can be computed from the energies using a sigmoid function $\sigma$:

$$\sigma(\log r_a(z', z)) := \sigma(\log r_a(f_\phi(x'), f_\phi(x))) = \sigma\left(\sum_i E_\theta(z'_i, a, z) - E_\theta(z'_i, a_0, z)\right).$$

Finally, we train our energy functions to match the observed ratios by training $|A| - 1$ classifiers computed by $\sigma(\log r_a(z', z))$. We use the transitions of other actions as negative samples and minimize the following binary cross-entropy loss:

$$\mathcal{L}_r(\theta, \phi) = \sum_{a' \in A} [a' = a] \log \sigma(\log r_a + \zeta_a) + [a' \neq a] \log(1 - \sigma(\log r_a + \zeta_a)); \tag{6}$$

where $[\cdot]$ is indicator functions that is 1 when the condition holds, and $\zeta_a := \log \frac{\pi(a|z)}{\pi(a_0|z)}$ are correction weights to account for the policy used to collect the data. In practice, we estimate the policy from the dataset and use the estimate to compute the loss. Finally, we minimize a weighted sum of these losses and use AdamW as our optimizer (Loshchilov & Hutter, 2019). Algorithm 1 formalizes the approach.

**Identifiability** The core assumption of ACF is that variables are independently controllable, that is, for every state variable $s_i$, there exists a context $s \in \mathcal{S}$ and action $a \in A$, where the action effect is sufficiently different from the natural dynamics of the variable ($a_0$ effect). The following theorem establishes identifiability of independently controllable factors if the solution found is sparse.

**Theorem 3.1** (Identifiability of the Independently Controllable Factors). *Let the learned encoder $f : X \to Z$ be a diffeomorphism. If the following conditions hold*

---

**Algorithm 1** Action Controllable Factorization

---

**Require:** Dataset $\mathcal{D} = \{(x, a, x')\}$, encoder $f_\phi$, set per-factor energy models $\{E_\theta^k\}_{k=1}^K$, policy $\pi_w$,
Learning rate $\alpha$, weights $\beta_r, \beta_{\text{fwd}}, \beta_{\text{inv}}, \beta_\pi$
1: **for** minibatch $\{(x^n, a^n, x'^n)\}_{n=1}^N \sim \mathcal{D}$ **do**
2:    Encode: $z^n \leftarrow f_\phi(x^n), \ z'^n \leftarrow f_\phi(x'^n)$
3:    Noise: $z^n \leftarrow z^n + \varepsilon^n, \ z'^n \leftarrow z'^n + \varepsilon'^n$
4:    Negatives: $\mathcal{N} = \{(z^i, a^j, z'^j) \mid i, j = 1, \ldots, N\}$
5:    Energies: $E_{ij}(a) = \sum_k E_\theta^k(z_k'^{\,j}, a, z^i) \ \forall i, j \in [N], k \in [K], a \in A$
6:    Policy logits: $\pi_{\text{logits}}^n = \pi_w(z^n) \ \forall n$
     {The diagonal values are the energy that correspond to real transitions}
7:    Ratios: $\log r_a^{nn} = E_{nn}(a) - E_{nn}(a_0)$
8:    Policy weights: $\zeta_a^n = \log \frac{\pi(a_n \mid z^n)}{\pi(a_0 \mid z^n)}$
9:    $\mathcal{L}_r = -\frac{1}{N} \sum_n \sum_a [a^n = a] \log \sigma \left( \log r_a^{nn} + \text{sg}\,(\zeta_a^n) \right) +$
        $[a^n \neq a] \log \left( 1 - \sigma \left( \log r_a^{nn} + \text{sg}\,(\zeta_a^n) \right) \right)$
10:    $\mathcal{L}_{\text{fwd}} = -\frac{1}{N} \sum_n \log \frac{e^{E_{nn}(a^n)}}{\sum_j e^{E_{nj}(a^n)}}$
11:    $\mathcal{L}_{\text{inv}} = -\frac{1}{N} \sum_n \log \frac{\pi(a^n \mid z^n) e^{E_{nn}(a_n)}}{\sum_{a'} \pi(a' \mid z^n) e^{E_{nn}(a')}}$
12:    $\mathcal{L}_\pi = \frac{1}{N} \sum_n -\log \frac{e^{\pi_{\text{logits}}^n[a^n]}}{\sum_{a'} e^{\pi_{\text{logits}}^n[a']}}$
13:    $\mathcal{L} = \beta_r \mathcal{L}_r + \beta_{\text{fwd}} \mathcal{L}_{\text{fwd}} + \beta_{\text{inv}} \mathcal{L}_{\text{inv}} + \beta_\pi \mathcal{L}_\pi$
14:    Update: $(\phi, \theta, w) \leftarrow \text{AdamW}((\phi, \theta, w), \alpha, \nabla \mathcal{L})$
15: **end for**

---

1. $\mathcal{S} \subset \mathbb{R}^K$ is connected and the unknown observation function $o : \mathcal{S} \to X$ is a diffeomorphism.

2. The action effects are **sufficiently different** from the natural dynamics. That is, there exists $i \in [K]$

$$\frac{\partial}{\partial s_i'} \frac{T_i(s_i' \mid s, a)}{T(s_i' \mid s, a_0)} \neq 0$$

   for $s \in \tilde{S} \subseteq \mathcal{S}$, almost surely. Moreover, there exists at least an action that affects each $s_i$ (independent controllability)

3. All energy function approximate the factor forward dynamics $E(z_i', a, z) \propto \log T(z_i' \mid z, a)$;

4. (**Sparsity**) The score differences (gradients of the energies)

$$\frac{\partial}{\partial z_i'} \Delta E_i^a = \frac{\partial}{\partial z_i'} \left[ E(z_i', a, z) - E(z_i', a_0, z) \right] \neq 0$$

   for at most one variable $j$ and all actions.

then, there exists a factor-wise diffeomorphism $h : \mathcal{S} \to Z$ between the underlying ground truth factors of variation $\mathcal{S}$ and the learned encoding $Z$

Similar arguments have been used to establish identifiability under action and state-dependency sparsity (Lachapelle et al., 2022; 2024) and under single-node interventions in causal representation learning (Varici et al., 2024). Theorem 3.2 can be viewed as a special case of these results adapted to the independently controllable factors setting[3]. This result further shows that independently controllable factors can be recovered when, in addition to certain regularity and variability conditions, the solution is sparse (Condition 4). We conjecture that the binary classifiers arising from the $\mathcal{L}_r$ loss promote sparsity by *competing* to capture what makes each action distinct with respect to both the natural dynamics and the other actions—namely, the specific factor influenced by the action. In the

---

[3]The proof is provided in Appendix A

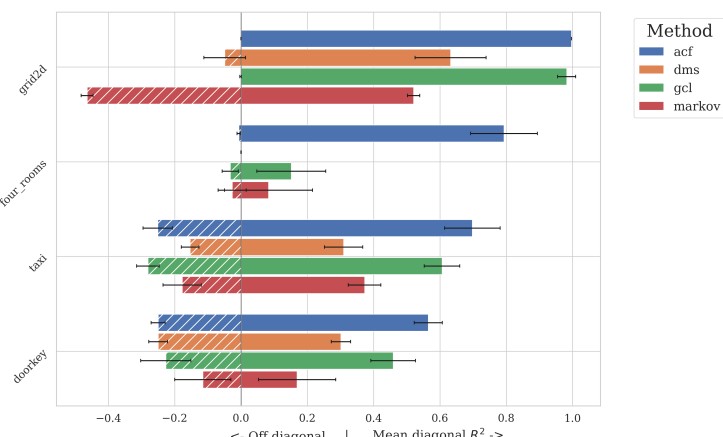

Figure 1: **Factorization metrics**. The left side bars show how much of the information is represented off the diagonal on average over all variables. The right side bars represent the mean diagonal value. Ideally, we would expect our $R^2$ matrices to be close to the identity: 0 on the left bar, 1 on the right bar. The error bars show the standard deviation over 5 independent seeds.

next section, we demonstrate empirically that ACF indeed identifies the independently controllable factors in practice.

## 4 EVALUATION

In this section, we empirically evaluate ACF in RL test domains directly from pixel observations. We consider the visual variation of the classical Taxi domain (Dietterich, 2000) and visual Minigrid environments (Chevalier-Boisvert et al., 2023): FourRooms (Sutton et al., 1999) and DoorKey[4]. We chose these domains because they allow easy access to the generating factors for evaluation and while these domains are simple from the perspective of learning a policy, they actually are challenging from the factorization problem perspective, as we will see in the quantitative results.

**Baselines** We consider GCL (Generalized Contrastive Learning; Hyvärinen et al. (2019)) that can be seen as a vanilla contrastive-based disentanglement algorithm, and DMS (Disentanglement via Mechanism Sparsity; Lachapelle et al. (2022)), a VAE-based (Kingma & Welling, 2014) method that explicitly maximizes sparsity in state dependencies and action effects to drive disentanglement. Moreover, we consider MSA (Markov State Abstractions; Allen et al. (2021)), a contrastive-based algorithm that leverages both forward and inverse dynamics to learn Markovian representations but does not explicitly optimize for disentanglement. It is important to notice that previous work has theoretically shown that methods that lack the correct inductive biases will converge to entangled representations almost surely (Locatello et al., 2019), therefore, we do not include these as baselines for our identification evaluation.

**Evaluation Protocol** To measure disentanglement, we consider test datasets of pairs of $\{(s^i, z^i)\}_i$ where $s$ is the ground truth representation and $z$ is the corresponding learned latent representation. Then, we fit factor-wise regressors (parameterized by feed-forward networks), $h_{ij}(z_i) \mapsto s_j$. The performance of $h_{ij}$ is limited by the amount of information $z_i$ contains about $s_j$, therefore we measure the quality of the learned regressor using the coefficient of determination $R^2$. Therefore, for each method we have a matrix $R^2$ (see Figure 2); this matrix would have 1 in the diagonal and low off-diagonal values if the ground truth variables were perfectly identified. We tune all methods via random search in their respective hyperparameter space and train 5 seeds for each method (see Appendix B).

**Quantitative results** Given a $R^2$ matrix, we search a permutation that maximizes the diagonal using the Hungarian algorithm (Kuhn, 1955). We then aggregate the matrices into two scores, the mean diagonal value, $\frac{1}{K} \sum_i^K R_{ii}^2$ and the mean maximum off-diagonal value $\frac{1}{K} \sum_i^K \max_{j \neq i} R_{ij}^2$. The

---

[4]We use Minigrid JAX (Bradbury et al., 2018) re-implementation (Pignatelli et al., 2024)

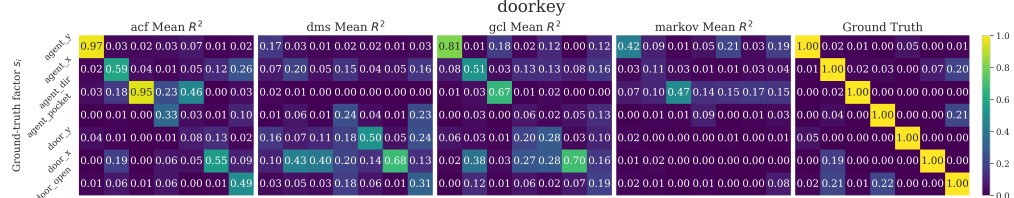

Figure 2: **Factorization matrices for DoorKey.** Mean $R^2$ matrices over 5 seeds.

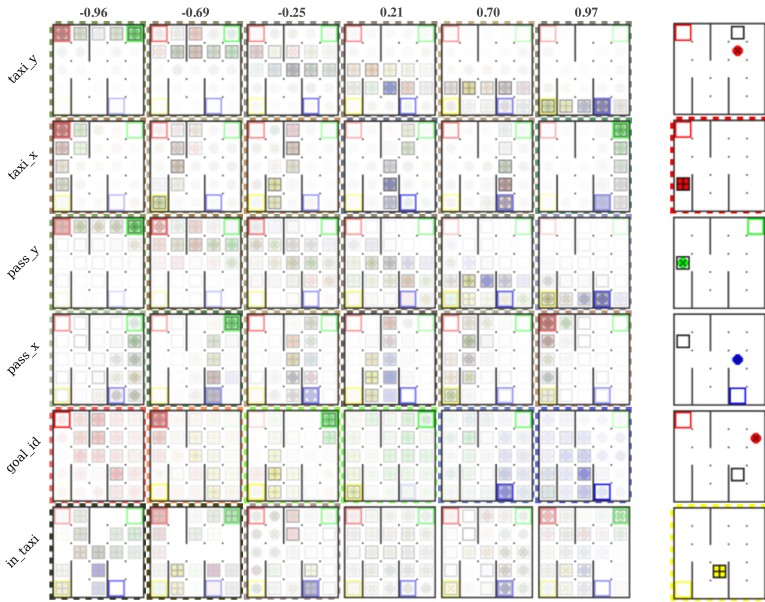

Figure 3: **Taxi latent traversals.** In this Taxi rendering, the taxi is represented by a hollow square, the passengers are circles with colors matching their goal positions. When a passenger is in the taxi, the border of the frame is highlighted with stripes. By varying the value of a latent variable (columns), we can see its effect on the mean observation. Each row represents different latent variables.

former measures how well a latent factor represents the ground truth factor, and the latter measures how much information is contained in the rest of the factors. Ideally, this would mean a score of 1 for the mean diagonal and 0 for the off diagonal if the identification is perfect and the factors are fully independent. However, this is only an upper bound on perfect performance in many environments; e.g. taxi and passenger's position are not fully independent because the passenger can only move if it moves with the taxi. Figure 1 shows the results for all methods and domains.

**The factor affected by an action depends on the current state $s$.** Grid2D and FourRooms have different factorizations: In grid2D the agent can move up, down, left and right and 2 dimensions are enough as controllable factors, but in FourRooms (Minigrid variant) the agent can rotate, move forward, backward, left or right and, hence, 3 factors are required. More importantly, the factor an action affects is *relative to the agent's orientation* and, this, change causes difficulties for all baseline methods. In particular, DMS, which assumes a global sparse graph, struggles to converge.

**Factors are not independent** In the Taxi domain, factorization is more challenging because the taxi's position and the passenger's location are inherently coupled; the passenger can move only if it moves with the taxi. Our method outperforms the baselines in this case. Figure 3 shows qualitatively the effect of traversing the identified passengers position variables.

**Identifying non-controllable variables** In the DoorKey domain, not all factors are controllable by the agent: the door's position is sampled at the start of each episode and kept fixed throughout the episode. Although the agent must perceive the door's location to open it, that factor need not be disentangled. In fact, as seen in Figure 2, the door $y$ coordinate is not identified. DMS, instead is able

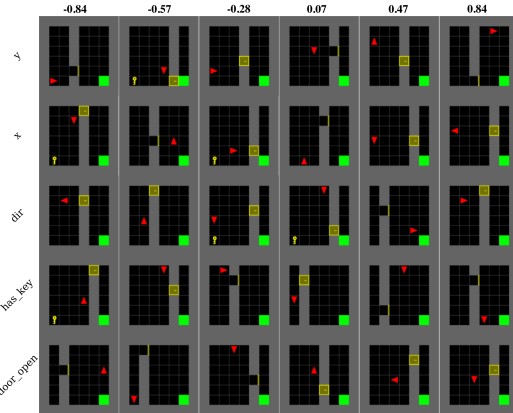

Figure 4: **DoorKey latent traversals**. For this domain, we show a random sample from observations that have a particular value of the latent dimension. We only show the controllable elements in DoorKey, that includes the agent position and orientation, the key and the door state. Different rows correspond to different latent variables and different columns represent different values for the corresponding latent variable.

to partially identify these variables because it's not constrained to controllable elements and uses the sparsity of the state dependencies.

**Ablation** We performed an ablation study in the Minigrid-Doorkey domain, the most challenging environment considered in the previous experiments. Table 1 shows that each loss term plays an important role in improving factorization. In addition, we evaluate the *Factored Markov* variant, which combines our unified factored energy parameterization with the MSA losses (forward and inverse). This variant achieves improved factorization compared to the original MSA, highlighting the importance of the our proposed parameterization.

| Experiment | Diag Score (Mean $\pm$ Std) $\uparrow$ | Off-Diag Score (Mean $\pm$ Std)$\downarrow$ |
|---|---|---|
| ACF (full) | **0.5650** $\pm$ 0.0423 | 0.2499 $\pm$ 0.0213 |
| Factored Markov | 0.4861 $\pm$ 0.0491 | 0.2635 $\pm$ 0.0339 |
| no fwd | 0.1987 $\pm$ 0.0588 | 0.1028 $\pm$ 0.0340 |
| no inv | 0.5294 $\pm$ 0.0274 | 0.1918 $\pm$ 0.0280 |
| no policy | 0.4630 $\pm$ 0.0694 | 0.2301 $\pm$ 0.0543 |
| no ratio | 0.5083 $\pm$ 0.0767 | 0.2353 $\pm$ 0.0352 |

Table 1: Ablation on Minigrid-Doorkey Enviroment over 5 seeds

## 5 RELATED WORKS

**Factored RL** There is a long history of leveraging the structure of FMDPs for efficient planning algorithms. By assuming that the structure of the MDP is known (i.e., the DBN), these algorithms exponentially reduce the size of the problem representation. Such algorithms include Structured Value Iteration algorithms (Boutilier & Dearden, 1996; Boutilier et al., 2000) and Structured Policy Iteration (Boutilier et al., 1995; Koller & Parr, 2000), and their extensions to linear approximation (Guestrin et al., 2003). In classical PAC model-based RL algorithms, the Factored $E^3$ algorithm (Kearns & Koller, 1999) extends the $E^3$ algorithm (Kearns & Singh, 2002) to the case where the DBN is known and an oracle factored planner is available. Guestrin et al. instantiate this algorithm and RMax Brafman & Tennenholtz (2002) using factored linear value iteration (Guestrin et al., 2002) as the planner. Algorithms such as SLF-RMax (Strehl et al., 2007), Met-RMax (Diuk et al., 2009) and SPITI (Degris et al., 2006) loosen the assumption of a known DBN structure and discover this structure online for discrete state spaces. Vigorito & Barto (2009) extends structure learning to continuous state and action spaces. Further theoretical work includes regret bounds for factored RL

in the episodic (Osband & Van Roy, 2014; Tian et al., 2020) and non-episodic settings (Xu & Tewari, 2020).

The discovery of the structure among the factored state variables has been used to do exploration (Seitzer et al., 2021; Wang et al., 2023). Closely related, in skill discovery, the factored structure can be exploited to generate signals that facilitate learning useful skills (Vigorito & Barto, 2010; Hu et al., 2024; Wang et al., 2024; Chuck et al., 2024; 2025). Moreover, leveraging the sparsity of edges in the DBN enables algorithms to learn modular world models that are robust (Wang et al., 2022b; Ke et al.). Counterfactual Data Augmentation (CODA) (Pitis et al., 2020; 2022) uses the *local* DBN structure to generate plausible transitions by recombining states based on conditional independence relations that hold locally—i.e., leverages local DBNs to have a nonparametric transition model. However, all of these require knowing the factored representation a priori for these methods to work.

Limited attempts exist to learn factorized representations in deep RL from high-dimensional observations. DARLA (Higgins et al., 2017b) leverages $\beta$-VAE representations for zero-shot generalization in multitask RL. Some works that try to learn factored world models include variational causal dynamics models (Lei et al., 2022) and provable factored RL (Misra et al., 2021). DenoisedMDP (Wang et al., 2022a; Liu et al., 2023) factorizes the state in four factors based on their relevance to reward and controllability. TED (Temporal Disentanglement; Dunion et al. (2023)) uses NCE from state transition samples to improve model-free agent robustness to correlated, irrelevant features, and CMID (Conditional Mutual Information for Disentanglement; Dunion et al. (2024)) uses causal, graphical conditions to infer the state factor from pixels. Perhaps, most similar to ours, is the work of Thomas et al. (2018) that proposes to learn independently controllable elements by learning policies that minimize the number of variables changed. However, they obtain limited success in fully aligning a 2D grid learned representation with the ground truth.

**Controllability-based Representations for RL** There is an extensive literature that focuses on the problem of reconstruction-free representations (Gelada et al., 2019; Zhang et al., 2021; Nguyen et al., 2021; Lee et al., 2020) and abstractions for RL. Moving away from reconstruction objectives allows more robust performance to temporally-correlated noise (Zhang et al., 2021; Wang et al., 2022a; Rudolph et al., 2024; Ortiz et al., 2024). Importantly, and related to our approach, controllability has been proven useful to learn deep representations that are sufficient for RL agents. Learning inverse models (Lamb et al., 2023; Allen et al., 2021; Yi et al., 2023; Rudolph et al., 2024) has proven useful to learn representations that capture the controllable components of the state. However, because inverse models are insufficient (Lamb et al., 2023; Allen et al., 2021), they have been complemented by multi-step inverse models (Lamb et al., 2023) and forward models (Allen et al., 2021). Factorizing controllable variables has been explored in the past by DenoisedMDP (Wang et al., 2022a), they assume known block factorization to separate variables across two axes (controllable/uncontrollable and reward relevant/irrelevant). Follow up work propose an identifiable approach to the DenoisedMDP idea (Liu et al., 2023). However, their approach does not encourage the disentanglement within the blocks. ACF, though only focuses on the controllable variables, is designed to factorize to a finer granularity.

**Disentanglement in Representation Learning** In representation learning (Bengio et al., 2013), disentanglement has been extensively studied (Schmidhuber, 1992; Higgins et al., 2017a; Burgess et al., 2018; Chen et al., 2018; Klindt et al., 2020) as a desirable characteristic for generalizable representations. However, a widely accepted definition of disentanglement does not yet exist and solving this problem without the right inductive biases is impossible (Locatello et al., 2019). Unsupervised approaches leverage the Variational Autoencoder (VAE; Kingma & Welling (2014)) to learn latent representations that have a factorized prior (Kim & Mnih, 2018), minimize total correlation (Chen et al., 2018), and leverage temporal relations (Klindt et al., 2020). An important formalization of disentanglement is Independent Component Analysis (ICA; Comon (1994)). In particular, non-linear ICA (Hyvärinen et al., 2023), where a set of source variables is entangled by an unknown non-linear function. Approaches in non-linear ICA include contrastive methods (Hyvärinen et al., 2019; Hyvärinen & Morioka, 2016), energy functions (Khemakhem et al., 2020b), quantized methods (Hsu et al., 2024a;b), VAEs (Khemakhem et al., 2020a; Klindt et al., 2020) and sparse graphical conditions such as DMS (Disentanglement via Mechanism Sparsity; Lachapelle et al. (2022)) Another approach is causal representation learning (Schölkopf et al., 2021). This tackles the problem of discovering the *causal* variables by leveraging data coming from interventional and observational distributions. In this problem, the variables are not assumed to be independent as in ICA. Simple methods assume having access about what variable was intervened (Lippe et al., 2022; 2023b; Locatello et al., 2020)

and assume binary interventions (Lippe et al., 2023a). Recent works establish identifiability results for linear mixing models (Squires et al., 2023), non-linear mixing (Ahuja et al., 2023; Buchholz et al., 2023; Zhang et al., 2023) and non-parametric in the case of unknown interventions (i.e., without labels of the intervened variable) (von Kügelgen et al., 2024; Varici et al., 2024). ACF can be interpreted as a special case where the agent's actions induce interventional distributions and the natural dynamics are simply the observational distributions.

## 6 DISCUSSION AND LIMITATIONS

ACF takes a step toward closing the factored-representation gap: the longstanding challenge of identifying independently controllable latent variables directly from high-dimensional observations. Our focus in this work is not on demonstrating downstream RL gains—prior work has already shown that when a factored representation is available, RL can benefit substantially—but rather on the foundational question of whether such factors can be reliably recovered from pixels. A key strength of ACF is that it does not assume intervention masks (Lippe et al., 2022; 2023b; Locatello et al., 2020), one-to-one mappings between actions and state variables (Lippe et al., 2023a), nor determinism or static-world dynamics (Thomas et al., 2018). Instead, ACF leverages actions as soft interventions and allows the effect of an action to depend on context, improving over prior disentanglement and controllable-state approaches. The simplicity of our evaluation domains is intentional: these environments provide accessible ground-truth factors, allowing us to directly evaluate whether the learned latent space matches the factored structure.

At the same time, important limitations remain. ACF assumes that the immediate effect of an action can be observed; thus long-term, delayed action effects still pose a challenge. Similarly, while nothing in the formulation requires discrete actions, continuous-control environments often exhibit precisely such delayed effects, and handling them requires additional considerations that we left for future work, as they deserve their own separate treatment. Finally, ACF assumes that the controllable factors are present in the observations; partial observability, occlusions, and nuisance variation require additional assumptions that are orthogonal to the identifiability question we study here.

These limitations outline clear directions for future work: integrating temporally extended actions or skills, extending ACF to partial observability, and incorporating the method into world-model or model-based RL frameworks where identified controllable variables can directly contribute to improved generalization and sample efficiency.

## 7 CONCLUSION

We introduced a new contrastive algorithm for learning a factored representation that recovers the independently controllable variables from high-dimensional observations. We use the fact that RL agents can act upon their environments and create discrepancies in the dynamics; contrasting those controlled dynamics to the natural order of things provides a signal for disentanglement that is relevant for factored RL. Moreover, we showed empirically that our method is able to recover the relevant controllable factors.

ACF shows that agent intervention and control over its environment is an important direction to achieve disentanglement. The converse is also relevant, a drive to discover the world structure should be explored as a new intrinsic signal for agent exploration, learning and skill discovery: *the agent needs to actively learn complex behavior and experiment to discover the structure of its environment*.

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

## A  THEORETICAL RESULTS

**Definition A.1** (Identifiability). An encoder $f : X \to Z$ identifies $\mathcal{S}$ if there exists a permutation $\pi$ and functions $h_i : \mathcal{S}_i \to Z_{\pi(i)}$ that are diffeomorphism such that $s_i = h(z_{\pi(i)})$. That is, each latent factor learned is equivalent to a ground truth factor up to a permutation.

**Lemma A.2** (*Local* Identifiability of the Independently Controllable Factors). *Let the learned encoder $f : X \to Z$ be a diffeomorphism. If the following conditions hold*

1. *The action effects are **sufficiently different** from the natural dynamics. That is, there exists $i \in [K]$*

$$\frac{\partial}{\partial s_i'} \frac{T_i(s_i' \mid s, a)}{T(s_i' \mid s, a_0)} \neq 0$$

   *for $s \in \tilde{S} \subseteq \mathcal{S}$, almost surely. Moreover, there exists at least an action that affects each $s_i$ (independently controllability)*

2. *All energy function approximate the factor forward dynamics $E(z_i', a, z) \propto \log T(z_i' \mid z, a)$;*

3. *(**Sparsity**) The learned score differences*

$$\frac{\partial}{\partial z_i'} \Delta E_i^a = \frac{\partial}{\partial z_i'} \left[ E(z_i', a, z) - E(z_i', a_0, z) \right] \neq 0$$

   *for at most one variable $j$.*

*then, there exists a factor-wise diffeomorphism $h : \mathcal{S} \to Z$ between the learned encoding $Z$ and the underlying ground truth factors of variation $\mathcal{S}$.*

*Proof.* Let $h$ be a diffeomorphism between $\mathcal{S}$ and $Z$. Given that the ground truth observation function is a diffeomorphism $o : \mathcal{S} \to X$ and, by assumption, the learned encoding is also a diffeomorphism. We can see that $h(s) = f(o(s))$. We need to prove that there exists a permutation $\pi : [K] \to [K]$ such that the permuted Jacobian of $h$, $P_\pi J_h$, is a diagonal matrix.

Given that $h$ is a diffeomorphism, $J_h$ exists.

Moreover, for each binary classifier, we know that at an optimum they converge to:

$$\log \frac{T(s' \mid s, a_n)}{T(s' \mid s, a_0)} = \sum_{l=1}^{K} E(z_l', a_n, z) - E(z_l', a_0, z) + C(z, a_n) \ \ \forall a_n \in A \setminus \{a_0\}; \qquad (7)$$

where $C(z, a)$ is a constant resulting from the normalization constants that are not estimated in ACF. By taking the gradient with respect to $s'$ using the chain rule, we get that

$$\nabla_{s'} \log \frac{T(s' \mid s, a_n)}{T(s' \mid s, a_0)} = \sum_{l=1}^{K} J_h^T(s') \nabla_{z'} \left[ E(z_l', a_n, z) - E(z_l', a_0, z) \right] \ \ \forall a_n \in A \setminus \{a_0\} \qquad (8)$$

Let $\rho(a_i) \mapsto [K]$ be the maps each action to its affected factor. Hence, by considering our sparse interaction model in Equation 1 we get that each classifier is a function of the variables affected by $a_n$. That is,

$$\nabla_{s'} \log \frac{T(s_{\rho(a_n)}' \mid s, a_n)}{T(s_{\rho(a_n)}' \mid s, a_0)} = \sum_{l=1}^{K} J_h^T(s') \nabla_{z'} \left[ E(z_l', a_n, z) - E(z_l', a_0, z) \right] \ \ \forall a_n \in A \setminus \{a_0\} \qquad (9)$$

$$= J_h^T \begin{bmatrix} \frac{\partial}{\partial z_1'}(E(z_1', a_n, z) - E(z_1', a_0, z)) \\ \vdots \\ \frac{\partial}{\partial z_K'}(E(z_K', a_n, z) - E(z_K', a_0, z)) \end{bmatrix} \qquad (10)$$

Moreover, consider a set of the actions $\bar{\mathcal{A}} \subseteq \mathcal{A} \setminus \{a_0\}$ such that each action affects one of the ground truth variables, that is, they are independently controllable.

We can write the above conditions in matrix form by stacking the gradients of all the actions in $\bar{\mathcal{A}}$.

Let $\Delta S(z' \mid z)$ be the matrices of learned score differences

$$[\Delta S(z' \mid z)]_{l,n} = \frac{\partial}{\partial z'_l} \left[ E(z'_l, a_n, z) - E(z'_l, a_0, z) \right]; \tag{11}$$

and $\Delta S(s' \mid s)$ be the matrices of score differences in $s'$.

$$[\Delta S(s' \mid s)]_{i,n} = \begin{cases} \frac{\partial}{\partial s'_i} \log \frac{T(s'_i | s, a_n)}{T(s'_i | s, a_0)} & \text{if } i = \rho(a_n) \\ 0 & \text{otherwise} \end{cases} \tag{12}$$

Hence, we can rewrite Equation 8 as

$$\Delta S(s' \mid s) = J_h^T(s') \Delta S(z' \mid z). \tag{13}$$

There exists $s$ such that all columns of $\Delta S(s' \mid s)$ has only one element different from zero and it is full rank because each factor is affected by at least one action. Moreover, given $J_h(s')$ is full rank because is a diffeomorphism and $\Delta S(z' \mid z)$ must also have exactly one element different from zero (sparsity condition).

Thus, $J_h(s')$ must have only one element different from zero per row. To see this, consider the $j$th column of $\Delta S(z' \mid z)_j = \beta e_r$ where $\beta \in \mathbb{R}$ and $r$ is the row different from zero. Hence,

$$\begin{aligned} \Delta S(s' \mid s)_j &= J_h^T(s') \Delta S(z' \mid z)_j; \\ &= \beta J_h^T(s') e_r; \\ &= \beta J_h^T(s')_{:,r}; \end{aligned}$$

and, therefore, $J_h(s')_r$ the $r$th column of the Jacobian must have one element different from zero. Therefore, $J_h(s')$ must be 1-sparse.

Finally, there exists a permutation $P(s')$ such that $J_h(s') = P(s')D(s')$ where $D(s')$ is a diagonal matrix. That is, there exists $h$ that is a factorwise transformation of $s$ up to a permutation.

$\square$

This lemma shows that the there exists a permutation in $s$ where action can have independent control over the factors. However, this does not guarantee the encoding is consistent because the permutation could be different in other parts of the space.

The following proposition establishes some conditions that guarantee identifiability globally.

**Proposition A.3** (Global Identifiability of Independently Controllable Factors). *Let the local conditions for identifiability hold. Moreover, assume that $\mathcal{S} \subset \mathbb{R}^K$ is connected. Then there exists a unique permutation $\pi$ for all $s \in \mathcal{S}$.*

*Proof.* Let $s_0 \in \mathcal{S}$ be a fix point. We know that the Jacobian $J_h(s_0) = P_\pi(s_0)D(s_0)$ because of local identifiability. Let $\pi_{s_0}$ be the permutation corresponding to the matrix $P_\pi(s_0)$.

Moreover, because $h$ is a diffeomorphism, we have that each derivative $h'_i(s)$ is continuous and non-vanishing.

Therefore, there exists a neighborhood $U$ such that $h_{\pi_{s_0}(i)}(s_{\pi_{s_0}(i)}) \neq 0$ for all $s \in U$.

Because of continuity of $J_h$, we must have that per each row it's nonzero element remains so and, similarly, for the zero elements of the row. Therefore, the permutation $\pi_{s_0} = \pi_s$ for all $s \in U$. This makes the permutation locally constant.

Finally, because there's a finite discrete number of permutations and $\mathcal{S}$ is connected, it implies that $\pi_s = \pi$ globally constant in $\mathcal{S}$.

$\square$

## B EXTENDED EMPIRICAL RESULTS

Anonymized code at https://anonymous.4open.science/r/factored_rl-EE0A.

### B.1 NETWORK ARCHITECTURES

All networks were implemented using JAX (Bradbury et al., 2018) and Flax NNX (Heek et al., 2024).

Table 2: All methods use the same Residual Convolutional architecture for encoder (and decoder, when required). All MLPs use SiLU activations. Latent encodings are Tanh to keep between $(-1, 1)$ except for DMS.

| Component | DoorKey(8x8) | Taxi | Grid2D | FourRooms |
|---|---|---|---|---|
| latent_dim $(d)$ | 7 | 6 | 2 | 5 |
| n_actions $n_a$ | 10 | 6 | 5 | 10 |
| ACF Energy[$\times d$] | $(d + n_a) \to 256 \to n_a$ | | | |
| ACF Inverse | $2d \to 128 \to n_a$ | | | |
| ACF Policy | $d \to 256 \to 256 \to n_a$ | | | |
| GCL Energy[$\times d$] | $(d + n_a) \to 128 \to 128 \to n_a$ | | | |
| DMS Transition[$\times d$] | $(d + n_a) \to 256 \to 1$ | | | |
| Markov Inverse | $2d \to 128 \to n_a$ | | | |
| Markov Ratio | $2d \to 128 \to 1$ | | | |

**Pixel-level Encoder & Decoder.**

We parameterize the residual blocks by doubling the depth of the output feature map until reaching a minimum resolution (`min_res`) (4 for all our experiments) starting from a minimum depth (24 for all our experiments). This is similar to the residual CNN used in Hafner et al. (2025). Table 2 show the details of the MLPs used.

- **Residual Encoder**:
    - Positional Embeddings ($x, y$ channels).
    - Cascade of downsampling ResidualBlocks: stride-2 $3 \times 3$ convolution $\to$ RMSNorm $\to$ SiLU, plus two $1 \times 1$ conv residual layers.
    - Flatten $\to$ 2-layer MLP (256$\to$256, SiLU, then Tanh) $\to$ `latent_dim`($d$)
- **Residual Decoder**:
    - MLP up-projection from `latent_dim` $\to$ min_res $\times$ min_res $\times$ $D$.
    - Stack of transposed ResidualBlocks (stride-2 conv$^T$, RMSNorm, SiLU, ...)
    - Central crop to $32 \times 32 \to$ Tanh activation.
- **MLPs** All MLPs have SiLU activations (Hendrycks & Gimpel, 2016).

### B.2 DOMAINS

In all domains we collected data by ensuring coverage of the state-action space. **Grid2D**

**Actions** No-op, Up, Down, Left, Right;

**State Space** Continuous 2D space

**Observations** $32 \times 32 \times 3$ pixel rendering.

**Taxi implementation in JAX.**

**Actions** No-op, Up, Down, Left, Right;

**State Space** $5 \times 5$, 1 passenger, 4 different goals positions;

**Observations** $32 \times 32$ RGB rendering.

**Minigrid-FourRooms & Minigrid-DoorKey(8x8)**

**Actions** No-op, Rotate clockwise, Rotate counterclockwise, Forward, Backward, Right, Left, Pickup, Open, Done;

**State Space** $16 \times 16$ grid (position) and orientation (North, South, East, West);

**Observation** RGB $32 \times 32$ rendering.

### B.3 HYPERPARAMETERS, TUNING AND COMPUTATIONAL RESOURCES

We tune the hyperparameters by random search allocating 50 samples to each method and each configuration run with 5 different seeds. Tables 3, 4, 5, and 6 show the details for the best results for all methods and domains. Each trial was run in a NVIDIA GeForce RTX3090 24GB. Each trial took 15min. All experiments used $150K$ transitions.

Table 3: Hyperparameters and coefficient weights for the ACF baseline across four domains.

| Hyperparameter | DoorKey–Uniform | Taxi | Grid2D | FourRooms |
|---|---|---|---|---|
| *Training* | | | | |
| batch_size | 128 | 128 | 128 | 128 |
| lr | $4.0966 \times 10^{-4}$ | $2.9126 \times 10^{-4}$ | $2.27497 \times 10^{-4}$ | $3.6392 \times 10^{-4}$ |
| epochs | 100 | 200 | 200 | 200 |
| *ACF Coefficients* | | | | |
| $\lambda_{\text{fwd}}$ | 97.815 | 31.444 | 95.395 | 40.736 |
| $\lambda_r$ | 25.623 | 5.018 | 48.560 | 16.963 |
| $\lambda_{\text{inv}}$ | 22.094 | 1.000 | 1.000 | 97.365 |
| $\lambda_\pi$ | 1.610 | 9.916 | 1.332 | 22.764 |

Table 4: Hyperparameters and coefficient weights for the GCL baseline across domains.

| Hyperparameter | DoorKey(8x8) | Taxi | Grid2D | FourRooms |
|---|---|---|---|---|
| *Training* | | | | |
| batch_size | 128 | 128 | 128 | 128 |
| lr | $2.0363 \times 10^{-4}$ | $4.4530 \times 10^{-4}$ | $1.6031 \times 10^{-4}$ | $2.7661 \times 10^{-5}$ |
| epochs | 100 | 200 | 200 | 200 |
| *GCL Coefficients* | | | | |
| classifier_coeff | 60.444 | 27.293 | 90.737 | 51.030 |
| recons_coeff | $9.26 \times 10^{-8}$ | $4.53 \times 10^{-10}$ | 0.193 | $3.50 \times 10^{-4}$ |

### B.4 DETAILED EMPIRICAL RESULTS

Detailed results for DoorKey (Figure **??**), Minigrid-FourRooms (Figure 6), Taxi (Figure 7b), and Grid2D (Figure 8).

Table 5: Hyperparameters and coefficient weights for the DMS baseline across domains.

| Hyperparameter / Coefficient | DoorKey(8×8) | Taxi | Grid2D | FourRooms |
|---|---|---|---|---|
| *Training* | | | | |
| batch_size | 128 | 128 | 128 | 128 |
| lr | $3.5332 \times 10^{-4}$ | $9.6832 \times 10^{-5}$ | $7.5007 \times 10^{-5}$ | $4.0218 \times 10^{-4}$ |
| epochs | 100 | 200 | 200 | 200 |
| *DMS Coefficients* | | | | |
| elbo_const | 4.737 | 67.349 | 42.948 | 5.225 |
| action_sparsity_const | 3.536 | 36.881 | 91.982 | 9.878 |
| state_sparsity_const | 2.557 | 8.024 | 1.000 | 6.091 |
| gumbel_temp | 7.417 | 1.000 | 6.360 | 3.465 |
| l2_reg_const | 0.0015 | 0.0023 | 0.2805 | 0.0060 |

Table 6: Hyperparameters and coefficient weights for the Markov baseline across domains.

| Hyperparameter / Coefficient | DoorKey(8×8) | Taxi | Grid2D | FourRooms |
|---|---|---|---|---|
| *Training* | | | | |
| batch_size | 128 | 128 | 128 | 128 |
| lr | $4.5536 \times 10^{-4}$ | $3.9622 \times 10^{-4}$ | $4.2654 \times 10^{-4}$ | $1.5854 \times 10^{-4}$ |
| epochs | 100 | 200 | 200 | 200 |
| *Markov Coefficients* | | | | |
| inverse_const | 6.009 | 66.916 | 78.480 | 9.472 |
| ratio_const | 8.519 | 42.518 | 1.000 | 0.311 |
| smoothness_const | 2.092 | 8.234 | 84.905 | 9.691 |

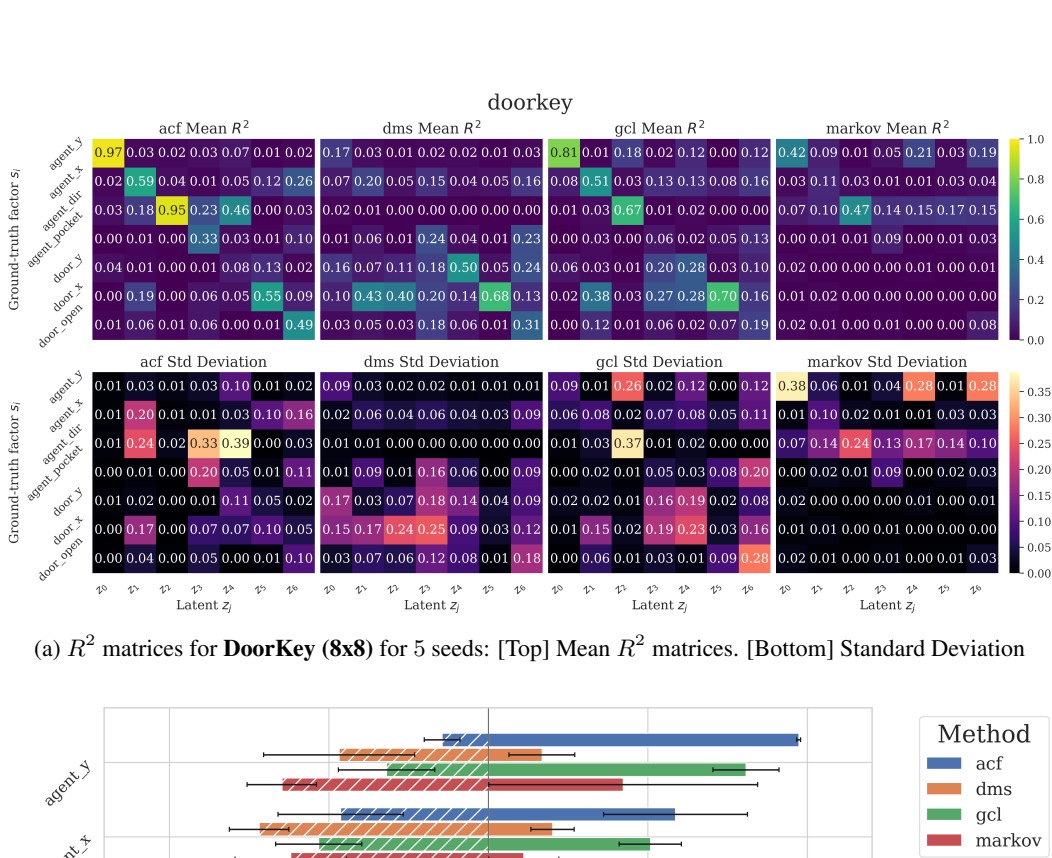

(a) $R^2$ matrices for **DoorKey (8x8)** for 5 seeds: [Top] Mean $R^2$ matrices. [Bottom] Standard Deviation

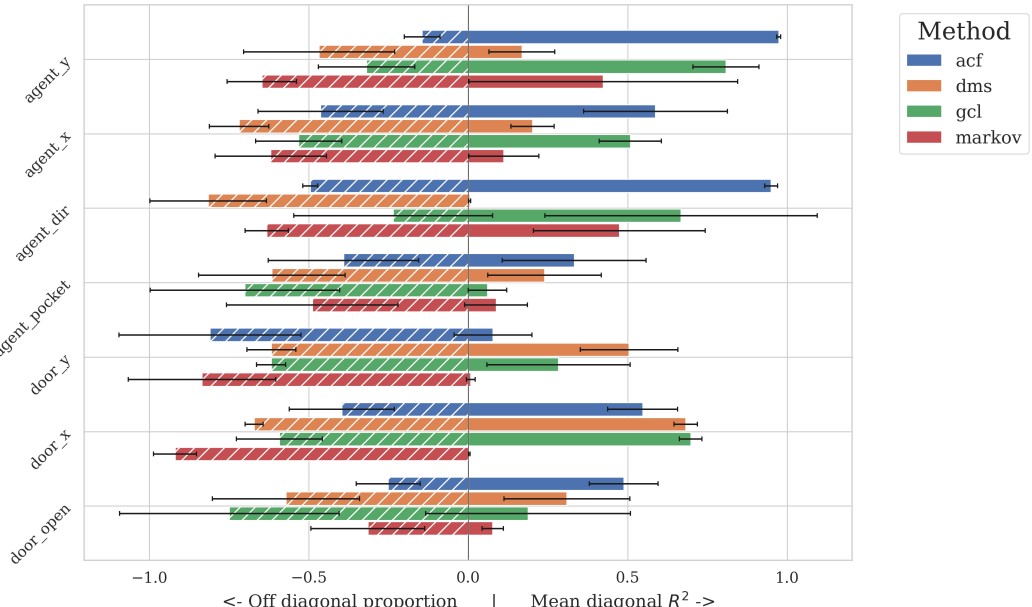

(b) Off diagonal proportion vs. Mean diagonal value per state

Figure 5: **DoorKey** Factorization Results

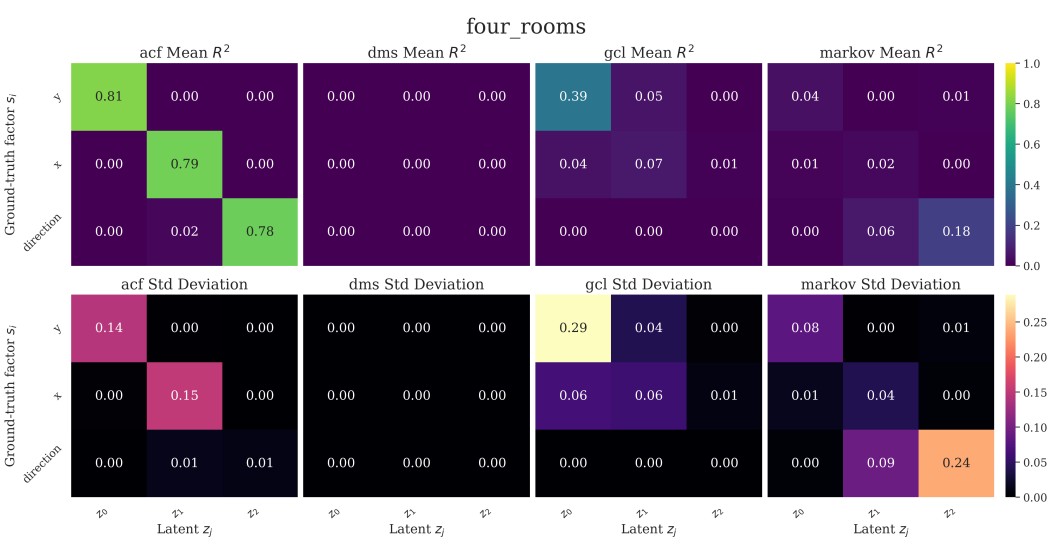

(a) $R^2$ matrices for **Minigrid-FourRooms** for 5 seeds: [Top] Mean $R^2$ matrices. [Bottom] Standard Deviation

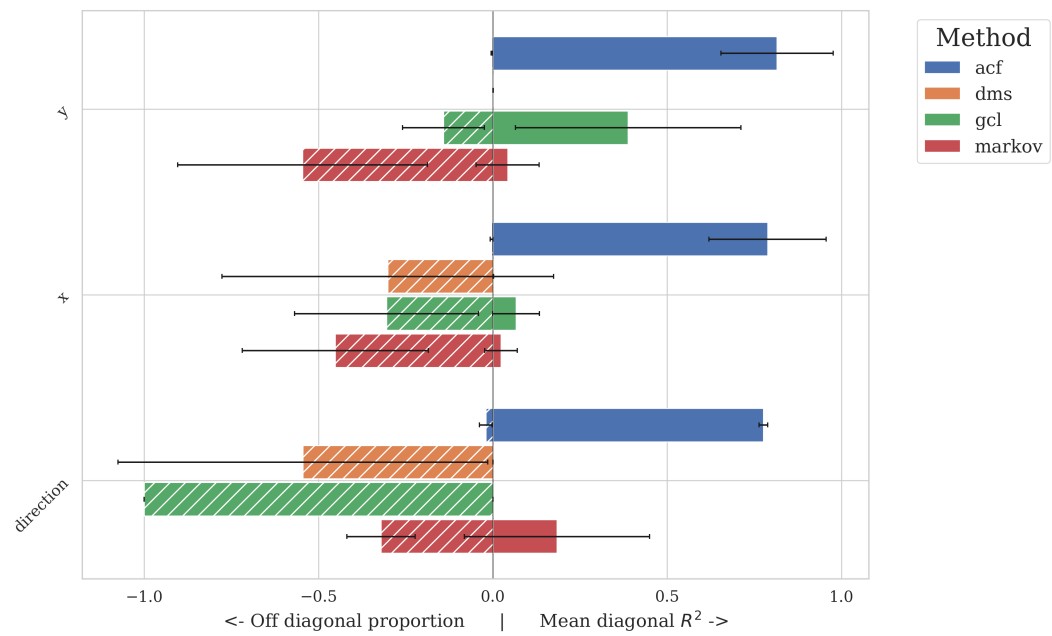

(b) Off diagonal proportion vs. Mean diagonal value per state

Figure 6: **Minigrid-FourRooms** Factorization Results

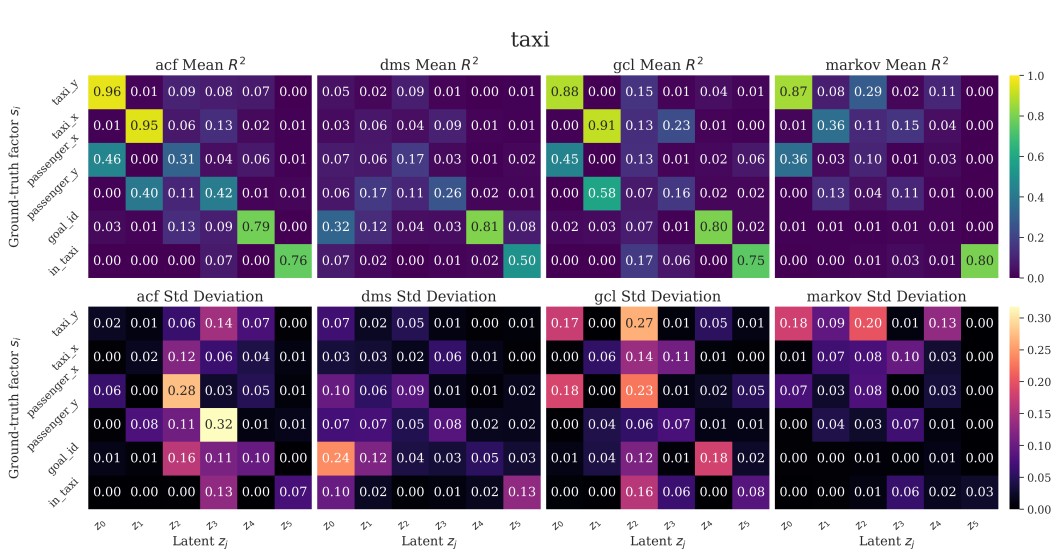

(a) $R^2$ matrices for **Taxi** for 5 seeds: [Top] Mean $R^2$ matrices. [Bottom] Standard Deviation

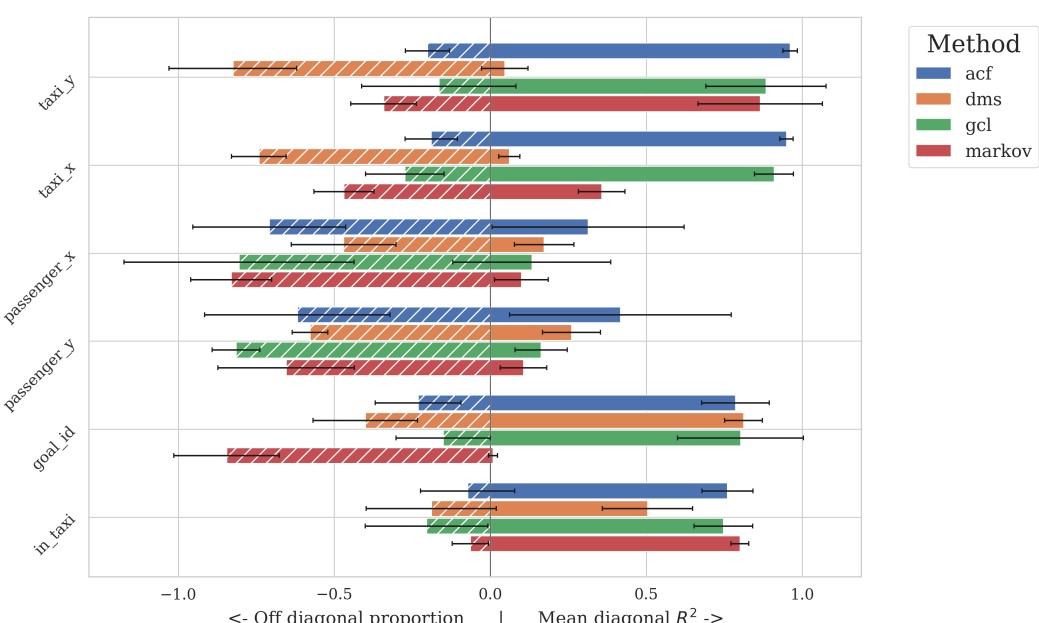

(b) Off diagonal proportion vs. Mean diagonal value per state

Figure 7: **Taxi** Factorization Results

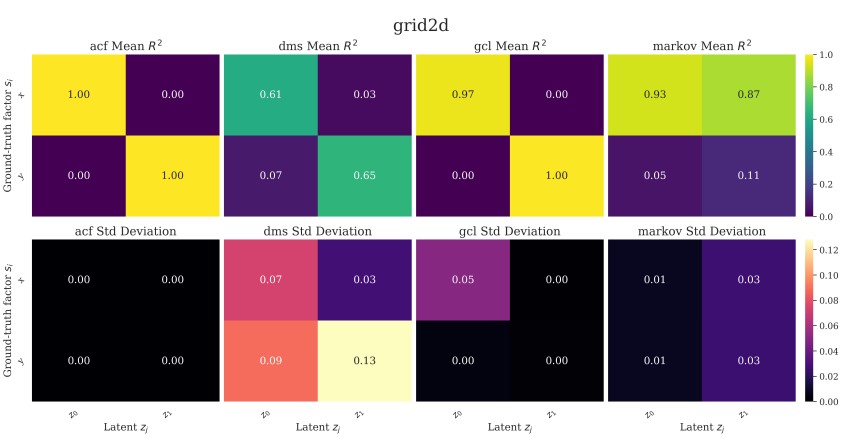

(a) $R^2$ matrices for **Grid2D** for 5 seeds: [Top] Mean $R^2$ matrices. [Bottom] Standard Deviation

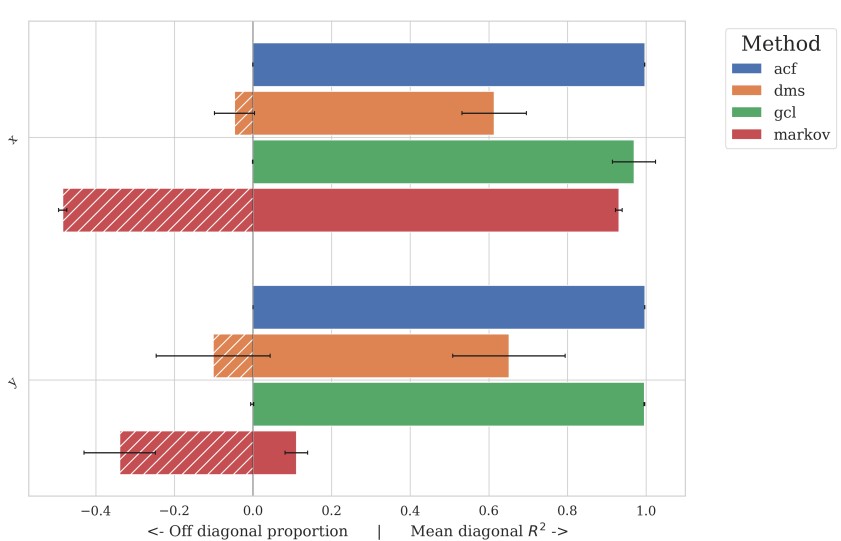

(b) Off diagonal proportion vs. Mean diagonal value per state

Figure 8: **Grid2D** Factorization Results