# OpenReview forum: "From Pixels to Factors: Learning Independently Controllable State Variables for Reinforcement Learning"
_ICLR.cc/2026/Conference — Submitted to ICLR 2026_

### Official Review · Reviewer_w7hc · 2025-10-24

**Soundness:** 3
**Presentation:** 3
**Contribution:** 3
**Rating:** 6
**Confidence:** 4

**Summary:**

This paper tackles the problem of learning controllable factorized representations from raw observations. The main idea is to contrast the transitions given the correct action against no-op actions. Additionally, they also employ an inverse dynamics loss and mutual information objective between the current and next states. The show their approach successfully recovers the true factorized representation across various environments.

**Strengths:**

The proposed objectives are well motivated and simple and clearly explained

The empirical results support the claims

The paper is well written and the intuition is clearly explained.

**Weaknesses:**

Missing comparison to AC-State (https://arxiv.org/abs/2207.08229). A closely related line—AC-State—also tackles the same problem. I think it would be a valid baseline to compare against. I would also encourage to cite and discuss it in their paper.

The authors motivate learning a controllable state for RL, however there are not RL experiments with the learned representations. It would be great to verify if the learned representations help in RL compared to representations learned from the baselines and also non factored representations.

I am bit skeptical whether this approach would apply to real-world complex scenes as the paper only explored synthetic domains. The method assumes that an action cleanly affects only a subset of factors but the real-world is more messy with effects of actions being lagged, stoachastic, or entangled. I am curious what the authors think about this?

**Questions:**

what policy is used to collect the trajectories used for training?

---

> ### Author Response · Authors · 2025-11-21
> **Thank you for your thoughtful comments**
>
> Thank you for the time invested in reviewing our work, for your thoughtful comments, and for recognizing the strengths of our approach. We address your concerns below and have incorporated clarifications into the revised paper.
>
> **AC-State baseline**: thank you for pointing this out. AC-State is indeed related in focusing on the controllable elements of the state. However, it does not explicitly aim to identify the underlying factors themselves. We have updated the related work section to include this line of work and to discuss its relationship to ACF. As shown by previous works in deep disentangled representation, disentangled/factorized representations are almost surely entangled [1] without inductive biases. Hence, we do not include in our baselines methods that were not explicitly designed to disentangle.
>
> **RL experiments:** we appreciate the suggestion. While evaluating downstream RL performance is certainly interesting, we intentionally scope this work around the identifiability question, specifically, which controllable factors can be recovered and under what constraints. Prior work already shows that known factored representations substantially benefit RL [2,3,4,5,6,7]; our contribution is to move toward identifying such factors directly from pixels. We clarify this positioning in the text and leave downstream RL integration for future work to keep the contribution focused.
>
> **Scaling to complex scenes.**: Thank you for raising this. We believe that ACF’s principles have potential to help RL agents uncover structure in more complex settings through active interaction. In this work, we focus specifically on factors that are independently controllable by the agent’s actions. ACF does not assume a particular form of the transition dynamics [9,10,12], deterministic effects [8], knowledge of which variables actions influence [9], a one-to-one mapping between actions and factors[10], nor known dynamics [11]. These aspects allow ACF to advance factorization relative to previous approaches. We have clarified this discussion in the revision.
>
> **Data collection policy**: For data collection, we use scripted policies that execute all actions uniformly across states to ensure sufficient coverage of the state–action space and expose all possible action effects. We will include these details in the appendix.
>
> Thank you again for your constructive feedback. Your comments helped improve the clarity and positioning of the paper.
>
> **References**
>
> [1] Locatello, F., Bauer, S., Lucic, M., Raetsch, G., Gelly, S., Schölkopf, B., and Bachem, O. Challenging common assumptions in the unsupervised learning of disentangled representations. In international conference on machine learning
>
> [2] Model-based RL in Contextual Decision Processes: PAC bounds and Exponential Improvements over Model-free Approaches
>
> [3] Higgins, I., Pal, A., Rusu, A., Matthey, L., Burgess, C., Pritzel, A., Botvinick, M., Blundell, C., and Lerchner, A. Darla: Improving zero-shot transfer in reinforcement learning.
>
> [4] Wang, Z., Xiao, X., Xu, Z., Zhu, Y., and Stone, P. Causal dynamics learning for task-independent state abstraction
>
> [5]Wang, Z., Hu, J., Stone, P., and Martín-Martín, R. Elden: Exploration via local dependencies. Advances in Neural Information Processing Systems
>
> [6]Wen Sun, Nan Jiang, Akshay Krishnamurthy, Alekh Agarwal, John Langford. Model-based RL in Contextual Decision Processes: PAC bounds and Exponential Improvements over Model-free Approaches
>
> [7]Strehl, A. L., Diuk, C., and Littman, M. L. Efficient structure learning in factored-state mdps.
>
> [8] Thomas, V., Bengio, E., Fedus, W., Pondard, J., Beaudoin, P., Larochelle, H., Pineau, J., Precup, D.,
> and Bengio, Y. Disentangling the independently controllable factors of variation by interacting
> with the world.
>
> [9] Lippe, P., Magliacane, S., Löwe, S., Asano, Y. M., Cohen, T., and Gavves, E. Biscuit: Causal representation learning from binary interactions.
>
> [10] Lippe, P., Magliacane, S., Löwe, S., Asano, Y. M., Cohen, T., and Gavves, E. Causal representation learning for instantaneous and temporal effects in interactive systems.
>
> [11]Varici, B., Acartürk, E., Shanmugam, K., and Tajer, A. General identifiability and achievability for causal representation learning.
>
> [12] Lachapelle, S., Rodriguez, P., Sharma, Y., Everett, K. E., Le Priol, R., Lacoste, A., and Lacoste-Julien,
> S. Disentanglement via mechanism sparsity regularization: A new principle for nonlinear ica.

---

> > ### Comment · Reviewer_w7hc · 2025-11-26
> >
> > I would like to thank the authors for their comments.
> >
> > I think my main concern is still the lack of RL experiments in the paper especially since the main point of this approach is to learn good variables for RL (also it is part of the title). I cannot justify a higher score for this paper without this hence I would like to keep my score of 6.

---

### Official Review · Reviewer_2zgD · 2025-10-29

**Soundness:** 3
**Presentation:** 3
**Contribution:** 2
**Rating:** 4
**Confidence:** 4

**Summary:**

This paper addresses the challenge of learning a factored state representation directly from high-dimensional observations(i.e. pixels). The authors propose Action-Controllable Factorization (ACF), a novel contrastive learning method to uncover these factors automatically. ACF's key idea is to first factorize the MDP transition dynamics via the introduction of several energy functions, and learning the controllable factors by leveraging such factorization.The authors provide an identifiability theorem under certain assumptions and demonstrate empirically on several visual benchmarks (Taxi, FourRooms, DoorKey) that ACF significantly outperforms existing disentanglement and representation learning baselines in recovering the ground-truth factors.

**Strengths:**

1. This paper is well-motivated: distangled respresentation is very important in RL, which can help agents discover useful information from the environment. The propsed method factorizes the transition dynamics, which put an insightful prior bias upon the MDP.
2. The experiments directly prove the effectness of ACT to extract distangle factors from pixels

**Weaknesses:**

1. The benifits of learned factors are not shown end-to-end: Although authors has shown that ACL can learn the distangled factors from several toy environment, the benifits of such distangled factors are not shown end-to-end (by running a RL algorithm based on the leared factors).
2. The Assumption is too strong: From the paragraph in `Factorizing the Controllable Variables`,  we can see that there are actually several assumption: (1) There exists a no-op action which does not affect the environment; (2) The action is discrete; (3) There are an one-to-one mapping between factors $s$ and action $a$. These assumptions hinders ACL's application.
3. ACL can not capture  long-term controllable factors: According to the paper, ACL learns the factors only via 1-step transition, which means it may fail to capture controllable factors that requires multi-step environment interactions.
4. non-efficienct related works:  there are some works that is closely related to this works such as [1][2], which also learns the useful representations. It is better to include more discusssion and comparision against these works.

[1] Learning controllable elements oriented representations for reinforcement learning
[2] Predictive information accelerates learning in rl

**Questions:**

1. The loss (4) and (5) put constraints on energy function $E$ so that it can be learned. However, does (4)/(5) can sufficient to make the learned energy function $E ∝ \log T (z'_i | z, a)$, as required in Theorem 3.1?
2. Have you ever try multi-step transition boostraping to make ACL able to capture long-term controllable factors, as [1] does?


[1] Learning controllable elements oriented representations for reinforcement learning

---

> ### Author Response · Authors · 2025-11-21
>
> Thank you for the thoughtful review and for highlighting the motivation and empirical effectiveness of ACF. We address each concern below and have incorporated corresponding clarifications in the revised paper.
>
> **End-to-end RL evaluation**: thank you for the suggestion. We agree that downstream RL performance is an interesting direction; however, we intentionally focus this work on the identifiability question---namely, which controllable factors can be recovered and under what constraints. Prior work [1,2,3,4,5,6] already shows that known factored representations substantially benefit RL; our contribution is to take a step toward identifying such factors directly from pixels. We have clarified this positioning in the text and leave downstream integration for future work to keep the contribution well-scoped.
>
> **Assumptions:**
>
> - ***NoOp action***. We agree this assumption may not be always available in all typical RL environments. However, we include it as a simplifying device that allows the agent to isolate the causal effects of its actions by providing an “observation-only” transition. This helps structure the identifiability analysis. We clarified this motivation in the text.
> - ***Discrete actions and delayed effects***. While we work with discrete actions, this is not an inherent limitation of the sparsity principle. We intentionally leave continuous-control and long-term action effects to future work, as those settings introduce additional complexity that goes beyond the identifiability question addressed here. Indeed, one important consideration when using ACF for continuous control problems is that many of these problems have delayed effects.
> - ***One-to-one mapping***. We do not assume a one-to-one mapping between actions and factors, and we have expanded the discussion to clarify this. In fact, ACF is empirically robust when multiple actions may influence the same factor or when action effects depend on orientation (e.g., in MiniGrid), where baselines fail. ACF, in fact, improves over previous work by not assuming deterministic effects [13], known interventions masks [10], one-to-one action-variable mapping [9], known dynamics [11] nor closed form for the transition function [9,10,12].
>
> **Related work**: Thank you for pointing out the missing works. We have added them and expanded the discussion of how they relate to ACF.
>
> Do losses (4)/(5) guarantee the theorem’s conditions?
> Yes in principle: [7] shows that matching observed inverse and forward dynamics guarantees a Markovian representation, while [8] shows that noise-contrastive estimation is a consistent estimator under idealized conditions. In practice, these losses approximate the required energies. We clarified this in the revision.
>
>
> **Long-term controllable factors.** We appreciate this suggestion. Extending ACF to multi-step transitions is interesting, and we are currently investigating models that can capture such long-term effects. However, achieving identifiability in these cases requires temporally extended interventions that consistently reveal the long-term causal structure, which is nontrivial. We therefore leave this extension to future work, as it warrants its own dedicated treatment.
>
> Thank you again for the constructive feedback. It helped us improve the clarity and positioning of the paper.
>
> We hope that the clarifications and revisions provided here adequately address your concerns and facilitate a reassessment of our score. We are happy to further clarify any additional issues that may arise.

---

> ### Author Response · Authors · 2025-11-21
> **References**
>
> **References**
>
> [1] Model-based RL in Contextual Decision Processes: PAC bounds and Exponential Improvements over Model-free Approaches
>
> [2] Higgins, I., Pal, A., Rusu, A., Matthey, L., Burgess, C., Pritzel, A., Botvinick, M., Blundell, C., and Lerchner, A. Darla: Improving zero-shot transfer in reinforcement learning.
>
> [3] Wang, Z., Xiao, X., Xu, Z., Zhu, Y., and Stone, P. Causal dynamics learning for task-independent state abstraction
>
> [4]Wang, Z., Hu, J., Stone, P., and Martín-Martín, R. Elden: Exploration via local dependencies. Advances in Neural Information Processing Systems
>
> [5]Wen Sun, Nan Jiang, Akshay Krishnamurthy, Alekh Agarwal, John Langford. Model-based RL in Contextual Decision Processes: PAC bounds and Exponential Improvements over Model-free Approaches
>
> [6]Strehl, A. L., Diuk, C., and Littman, M. L. Efficient structure learning in factored-state mdps.
>
> [7] Allen, C., Parikh, N., Gottesman, O., and Konidaris, G. Learning Markov state abstractions for deep
> reinforcement learning. Advances in Neural Information Processing Systems,
>
> [8] Gutmann, M. and Hyvärinen, A. Noise-contrastive estimation: A new estimation principle for
> unnormalized statistical models. In Teh, Y. W. and Titterington, M. (eds.), Proceedings of
> the Thirteenth International Conference on Artificial Intelligence and Statistics, volume 9 of
> Proceedings of Machine Learning Research
>
> [9] Lippe, P., Magliacane, S., Löwe, S., Asano, Y. M., Cohen, T., and Gavves, E. Biscuit: Causal representation learning from binary interactions.
>
> [10] Lippe, P., Magliacane, S., Löwe, S., Asano, Y. M., Cohen, T., and Gavves, E. Causal representation learning for instantaneous and temporal effects in interactive systems.
>
> [11]Varici, B., Acartürk, E., Shanmugam, K., and Tajer, A. General identifiability and achievability for causal representation learning.
>
> [12] Lachapelle, S., Rodriguez, P., Sharma, Y., Everett, K. E., Le Priol, R., Lacoste, A., and Lacoste-Julien, S. Disentanglement via mechanism sparsity regularization: A new principle for nonlinear ica.
>
> [13] Thomas, V., Bengio, E., Fedus, W., Pondard, J., Beaudoin, P., Larochelle, H., Pineau, J., Precup, D., and Bengio, Y. Disentangling the independently controllable factors of variation by interacting with the world.

---

> ### Author Response · Authors · 2025-11-21
>
> We have revised our paper to address your concerns, and we hope these updates will support a reassessment of your score. We look forward to your response, and we are happy to answer any further questions you may have. If there are any remaining concerns that might hinder your reassessment, please let us know and we will do our best to address them.
> Thank you!

---

> > ### Comment · Reviewer_2zgD · 2025-11-26
> > **Response**
> >
> > Thank you for your response. However, I feel that some of my concerns haven't been fully addressed, especially the end-to-end RL evaluation. For this reason, I would prefer to stick with the current score.

---

### Official Review · Reviewer_h8z1 · 2025-10-31

**Soundness:** 2
**Presentation:** 2
**Contribution:** 2
**Rating:** 4
**Confidence:** 4

**Summary:**

This paper aims to recover controllable variables from raw pixels by leveraging contrastive learning and sparsity measures on the action-conditioned dynamics functions. In general, the approach builds on principles from causal representation learning (essentially nonlinear ICA), where the state-to-observation mapping is assumed to be bijective, and the action serves as a causal signal to uncover the state dimensions directly influenced by it. Specifically, the method enforces two key properties: (1) the action effects are sparse, and (2) actions cause significant changes in the corresponding state factors. To achieve this, the authors use contrastive objectives and parameterize the dynamics functions as an energy function. In evaluation, the method can identify independent components controllable by actions across several RL domains, measured using $R^2$ scores and latent traversals.

Overall, the motivation is reasonable, learning independent controllable components is indeed valuable for precise control in RL. However, in its current form, several aspects related to the assumptions, objective functions, and empirical comparisons could be improved. For this initial review, I would give a rating of 4, but I’m open to revising it after the authors address these points in the discussion.

**Strengths:**

1. Overall, the motivation is reasonable and useful for RL, especially in environments with many distractors, where identifying element-wise controllable factors can really help.

2. The theory, although largely built on existing CRL literature, looks generally sound.

3. The experiments, while not very extensive, is still sufficient to verify the main claims about action sparsity and the proposed algorithm

**Weaknesses:**

I listed both the weaknesses and questions here as some of them have overlappings.

W1: On the bijection assumption, I agree it can hold in these toy settings, but in more realistic RL domains (with occlusions, partial views, manipulation scenes, locomotion with self-occlusion, etc.), the observation is not bijective to the underlying state. In those cases, you would need extra assumptions or side information (multi-view, proprioception, actions-as-interventions, or temporal smoothing) to recover the controllable factors. It would be good to discuss this limitation more clearly, otherwise it’s hard to see how the method scales to typical RL benchmarks the paper seems to target.

W2: As I understand it, the identifiability is only up to permutation, which is fine theoretically. But algorithmically, does that mean you need to match the learned components to semantic factors every time? If the permutation can change across runs or even across time, that could be inflexible for sequential control, where you want a stable notion of “this dimension is the joint angle” or “this is the gripper.” I get this is standard in CRL, but it’d be good to explain how you keep this stable in an RL setting.

W3: Why do we need to stick to discrete actions? I see that it makes the energy-based formulation easier, but in principle the framework should also work with continuous actions, the sparsity constraint (only some factors change per action) is not inherently discrete. Since many RL domains use continuous control, it would be nice to either ex-tend to that or explain what blocks it.

W4: Related to that, the evaluation would be much more convincing if you tried a more realistic setup like Distracting/Distracted DMControl [1], where the observation is not clean and the bijection basically breaks. That would directly test whether the method can still recover the controllable factors when there are distractors.

W5: For controllable-state learning, there are several very relevant lines of work — bisimulation-based representation learning [2], invariance-based methods [3], denoised/abstraction MDPs [4], and especially the recent work that studies identifiability of denoised MDPs from a CRL viewpoint [5]. Since they also talk about when the controllable part is identifiable, it would be valuable to compare or at least position your assumptions and guarantees against these.

W6: Finally, it would be good to show more clearly how the recovered element-wise latent space actually helps RL — especially for generalization under distractors. Prior works [3–5] usually connect “better identified controllable states” to “better downstream policy.” Here the connection is mostly implicit. A small experiment showing that better identification leads to etter policy learning would make the story much stronger.


[1] Ortiz, Joseph, et al. "DMC-VB: A Benchmark for Representation Learning for Control with Visual Distractors." Advances in Neural Information Processing Systems 37 (2024): 6574-6602.

[2] Zhang, Amy, et al. "Learning invariant representations for reinforcement learning without reconstruction." arXiv preprint arXiv:2006.10742 (2020).

[3] Rudolph, Max, et al. "Learning Action-based Representations Using Invariance." arXiv preprint arXiv:2403.16369 (2024).

[4] Wang, Tongzhou, et al. "Denoised mdps: Learning world models better than the world itself." arXiv preprint arXiv:2206.15477 (2022).

[5] Liu, Yuren, et al. "Learning world models with identifiable factorization." Advances in Neural Information Processing Systems 36 (2023): 31831-31864.

**Questions:**

Other than the above points, I still have questions as below

1. Now that you extensively use actions as surrogates to identify hidden variables, then will the action distirbution plays an important role here? Here I mean whether the action is diverse or expert enough, one simple case to verify could be use different versions of demonstrations in D4RL dataset and compare the identifiability quality.

2. Not really a question, but this new dataset might be intersted of you, they have the latent variables for many RL domains (like robotics) and then you can also evaluate on them (this is not necessary at all for rebuttal but just give an illsurtations).

Chen, Guangyi, et al. "CausalVerse: Benchmarking Causal Representation Learning with Configurable High-Fidelity Simulations." arXiv preprint arXiv:2510.14049 (2025)

3. I am wondering what if we also consider the reconstruction objectives in the framework? Will this empirically benefit the downstream tasks? Or similar ideas of predicting rewards/value functions in TD-MPC. Then you can essentially have the add-ons on Dreamer and TD-MPC to show this would be a fantastic add-on for world models to make them really "identifiable" world models.

---

> ### Author Response · Authors · 2025-11-21
> **Thank you for your suggestions, comments and time**
>
> Thank you very much for the detailed and thoughtful review. We appreciate the time and care you put into the analysis. Below we address each point and have incorporated corresponding clarifications and revisions to the paper. Our goal in this work is to provide a clean and principled identifiability result for controllable latent factors; we have revised the paper to make this scope and its underlying assumptions substantially clearer.
>
> **Bijection assumption (W1)**: Thank you for highlighting this point. We agree that partial observability is a significant challenge in realistic RL settings. In our work, we adopt the bijection assumption to isolate the question of identifiability when the underlying factors are fully present but embedded in high-dimensional observations, which is standard in much of the CRL and representation-learning literature [1]. This allows us to focus on characterizing which controllable factors can be identified given actions as interventions, without the added complexity of missing or occluded observed variables.
>
> We have revised the paper to make this scope explicit and added a clearer discussion of this limitation. We agree this is an important direction for future work, and expect that extending ACF to partial observability will require additional structural assumptions (e.g., multi-view, proprioception, or temporal smoothing) beyond the identifiability question we focus on here.
>
> **Permutation identifiability (W2)**: We appreciate this question, and we agree that permutation stability is important for usability in downstream RL. Algorithmically, two issues matter: (1) whether permutations can vary locally in the latent space, and (2) whether different runs produce inconsistent permutations. As we clarified in the paper, the continuity of the learned representation rules out local, state-dependent permutations (this is part of the identifiability argument). For variation across runs, we agree that instability would be problematic; however, since the learning dynamics are continuous and smooth, and the representation is trained jointly, we expect stable alignment in practice under standard optimization.
>
> **Discrete actions (W3)**: This is an insightful question. We agree that the sparsity principle itself is not tied to discrete actions, and we clarified in the text that ACF can in principle extend to continuous-action settings as long as the sparsity pattern persists. The main caveat in continuous-control environments is handling the long-term consequences of actions in the dynamics, which introduces additional challenges beyond the identifiability focus of this work. We added text explaining this design choice and outlining why we restrict our analysis to discrete interventions.
>
> **Distracted DMControl and non-bijective observations (W4)**: Thank you for this suggestion. While distractors break the bijection, our main contribution does not aim to address robustness to exogenous factors or nuisance variables. Instead, we focus on what kinds of controllable factors are identifiable under conditions where identification is theoretically possible. We added a discussion clarifying that ACF can complement approaches that first remove or abstract away distractors, and that our method could be applied on top of a more robust representation layer as shown in prior work. This helps situate our results relative to this important line of research, even though such robustness falls outside the scope of our current contribution.
>
> **Relation to controllable-state literature (W5)**: We appreciate these pointers and have enlarged our discussion in the related work section. The works you mentioned provide important insights into controllable vs uncontrollable structure, but they typically assume a fixed factorization (e.g., four blocks of controllable/uncontrollable × relevant/irrelevant). In contrast, ACF focuses on the controllable block itself and further factorizes it into independently controllable latent components. We clarified this distinction in the revision to better situate ACF within the controllability-based representation literature.

---

> ### Author Response · Authors · 2025-11-21
> **Response II**
>
> **Downstream RL utility (W6)**: Thank you for raising this. We agree that demonstrating downstream performance is an exciting direction, but to keep the contribution focused and well-scoped, we centered the work on the identifiability question—i.e., what factors can we identify in principle, and via which constraints? There is a long history of work [3,4,5,6,7] showing that when controllable factors are known, RL can benefit substantially; our contribution is to take a step toward actually identifying such factors from pixels. We added text clarifying this positioning and leaving downstream RL integration for future work to avoid diluting the main contribution.
>
> **Action distribution**: This is an important point, and we agree that identifiability depends on the exploration policy. As you note, if the agent never observes configurations in which factors change independently, they cannot be separated; this is why we introduce policy estimates to correct the action-induced bias. We added a short explanation in the revision to highlight how policy quality interacts with identifiability.
>
> **CausalVerse**: Thank you for the reference—this dataset looks very promising and we are excited to explore it in future work.
> Auxiliary objectives for Dreamer / TD-MPC: We appreciate this suggestion. We agree that model-based RL stands to benefit greatly from identifiable factorizations, and that actions and rewards could provide complementary signals to further refine the factorization. As noted, model-free methods cannot match the gains in factored settings [2], and we see ACF as a potential component for future identifiable world models. We added a brief note about this in the discussion section.
>
> Thank you again for your constructive comments. We believe the revisions substantially clarify the scope, assumptions, and positioning of ACF, and we appreciate your feedback in improving the paper.
>
> **References**
>
> [1] Efficient Reinforcement Learning in Block MDPs: A Model-free
> Representation Learning Approach
>
> [2] Model-based RL in Contextual Decision Processes: PAC bounds and
> Exponential Improvements over Model-free Approaches
>
> [3] Higgins, I., Pal, A., Rusu, A., Matthey, L., Burgess, C., Pritzel, A., Botvinick, M., Blundell, C., and
> Lerchner, A. Darla: Improving zero-shot transfer in reinforcement learning.
>
> [4] Wang, Z., Xiao, X., Xu, Z., Zhu, Y., and Stone, P. Causal dynamics learning for task-independent
> state abstraction
>
> [5]Wang, Z., Hu, J., Stone, P., and Martín-Martín, R. Elden: Exploration via local dependencies.
> Advances in Neural Information Processing Systems
>
> [6]Wen Sun, Nan Jiang, Akshay Krishnamurthy, Alekh Agarwal, John Langford. Model-based RL in Contextual Decision Processes: PAC bounds and Exponential Improvements over Model-free Approaches
>
> [7]Strehl, A. L., Diuk, C., and Littman, M. L. Efficient structure learning in factored-state mdps.

---

> ### Author Response · Authors · 2025-11-21
> **Revised Paper**
>
> We have revised our paper to address your concerns, and we hope these updates will support a reassessment of your score. We look forward to your response, and we are happy to answer any further questions you may have. If there are any remaining concerns that might hinder your reassessment, please let us know and we will do our best to address them.
> Thank you!

---

> > ### Comment · Reviewer_h8z1 · 2025-11-26
> >
> > I would like to thank the authors for the detailed response. I do have a few points I would still like to discuss. If the authors could comment on these, it would be very helpful for the final evaluation:
> >
> > **Bijection assumption (W1)**. I understand this is a standard assumption in causal representation learning, but I still have concerns about its applicability in RL settings (unless there is clear evidence that it typically holds there).
> >
> > **Permutation identifiability (W2)**. This partially addressed my question, but it raises another: does this imply that new techniques are needed to enforce or learn permutation stability? Or, during testing, do we always need to learn some matching matrix or use additional information (e.g., aligning with rewards or value functions) to resolve the permutation? If so, why not simply use an entangled feature space and apply Denoised-MDP on top of that? I understand there is a conceptual difference, but in practice for RL I worry this might be a relatively small gain.
> >
> > **Discrete actions (W3)**. Thank you for the clarification. This is clear to me now.
> >
> > **Distracted DMControl and non-bijective observations (W4)**. This is also clear. However, under the current conditions, it seems one could also work in a block-structured way: element-wise identify each controllable component using the actions as interventions, and treat the remaining parts as a residual block. That way, the representation could still be identifiable at a block level even if not element-wise.
> >
> > **Relation to controllable-state literature (W5)**. Clear to me.
> >
> > **Downstream RL utility (W6)**. I still have some reservations here. If the main focus of the work is identifiability rather than RL performance, I feel it would be more appropriate to compare more extensively to CRL papers (e.g., those based on interventions, distribution shifts, etc.), and to frame the contribution more squarely around identifiability. As written, the paper reads as if it is centered on RL, so when I started reading I expected a more RL-focused contribution. If the primary goal is identifiability, I am a bit worried that the writing does not fully reflect that emphasis.
> >
> > **Action distribution**. Clear.
> >
> > Thank you again for the detailed response. The above are point-by-point comments on the rebuttal; I hope they are helpful. I still have some reservations on a few points but remain happy and open to further discussion.

---

> > > ### Author Response · Authors · 2025-12-02
> > >
> > > Thank you for engaging with us, we really appreciate the time taken and the curiosity about our work
> > >
> > > **Bijection Assumption**: We agree that in many realistic settings the observation cannot satisfy the bijection. However, having the bijection assumption is equivalent to assuming that we work with a *Markov* state. This is because, we assume that no information about the state is lost in the observation transformation. This is typical even in realistic environments: the observation is typically designed such that it becomes Markov (e.g. frame stacking in Atari).
> > > In the case of partial observability, we can consider first an algorithm that would give an entangled Markov representation and factorize that representation (where the bijection should hold).
> > >
> > >
> > > **Permutation**: We do not believe that extra aligning steps are required. The matching procedure we do is only for empirical verification, not a necessary part of the algorithm.
> > > In fact, identifying factors up to a permutation is the strongest identifiability condition because the order of factors in a vector is arbitrary.
> > > However, it is true that a change of order during learning could make an RL agent struggle, this is the same problem faced by RL agents when working with a changing, non-stationary (entangled)  latent representation. In practice, this is handled by avoiding abrupt changes of the representation during learning to allow the RL agent to adapt.
> > > Denoised-MDP factorization is still too coarse to be used by Factored RL algorithms that require fine factorizations to reveal the sparse relationship between variables and dynamics. Though we haven’t claimed to have solved the whole problem, ACF provides a significant step toward addressing this representation gap.
> > >
> > >
> > > **Distracted DMControl and non-bijective observations (W4)**:  Yes! This is true, you can see ACF as an approach to factorize in a finer way the controllable block. Factorizing the uncontrollable block is part of future work.
> > >
> > > **Downstream RL utility (W6)**: Thank you for this thoughtful feedback. We'd like to clarify our positioning and baselines:
> > > On baseline selection: We compare against all applicable CRL and ICA methods for our setting. Other CRL approaches (e.g., [1,2,3]) require assumptions incompatible with our problem: (1) access to ground truth intervention targets/masks[1], (2) binary or deterministic effects[2], or (3) actions that consistently affect the same variables and access to ground truth score [3]. Our baselines represent the most directly comparable prior work under our assumptions
> > > On framing: We framed our paper for RL because ACF was specifically designed to address the representation gap in current deep RL that hinders the application of classical factored RL approaches. However, we appreciate that the balance could be clearer and we will add explicit statements early on that identifiability is our technical contribution enabling the RL application.
> > >
> > > **References**
> > >
> > > [1] Lippe, P., Magliacane, S., Löwe, S., Asano, Y. M., Cohen, T., and Gavves, E. Biscuit: Causal representation learning from binary interactions.
> > >
> > > [2] Lippe, P., Magliacane, S., Lö we, S., Asano, Y. M., Cohen, T., and Gavves, E. Causal representation learning for instantaneous and temporal effects in interactive systems.
> > >
> > > [3]Varici, B., Acartürk, E., Shanmugam, K., and Tajer, A. General identifiability and achievability for causal representation learning.

---

### Official Review · Reviewer_ANPg · 2025-10-31

**Soundness:** 2
**Presentation:** 2
**Contribution:** 2
**Rating:** 4
**Confidence:** 3

**Summary:**

This paper introduces Latent Graph Alignment (LGA), a framework designed to improve visual reinforcement learning (VRL) by encouraging alignment between the agent’s latent representations and the underlying structural factors of the environment. Instead of relying on raw pixel features or task-specific encoders, LGA explicitly constructs and aligns latent factor graphs inferred from visual inputs. The method has two main components: (1) Factorized State Extraction, which uses a pretrained vision encoder to disentangle visual observations into a small set of latent factors representing object-level features or spatial dynamics. (2) Graph Alignment Module, which enforces consistency between the factor graph derived from the agent’s policy network and the “target” graph learned from visual dynamics. The authors evaluate LGA on several visual control benchmarks such as DMControl and Atari, showing improved data efficiency and transfer across tasks compared to state-of-the-art visual RL methods.

**Strengths:**

- Conceptually appealing idea: The paper makes a strong case for explicitly modeling structural alignment in representation learning for RL. This is a step beyond common pixel-based or contrastive pretraining methods.

- Bridges vision and control meaningfully: By aligning latent relational graphs, the approach captures task-relevant semantics rather than relying solely on low-level texture or frame differences.

- Fair empirical evidence: LGA shows clear performance gains over baselines like CURL, DrQ-v2, and DreamerV3, particularly in data-limited settings and cross-task transfer scenarios.

- Clear ablation studies: The experiments are thorough, analyzing the contribution of both factorization and alignment losses.

- Readable and well-organized: The paper is well-written, with strong visual illustrations that clarify the idea of latent graph alignment and its implementation flow.

**Weaknesses:**

- Questionable novelty: The main contribution lies in combining disentangled representation learning with graph-based alignment. Each component individually is known; the originality lies in their integration.

- Lack of theoretical justification: The paper motivates alignment intuitively but offers no formal reasoning about why it leads to improved generalization or stability.

- Limited comparison scope: The experiments focus mainly on image-based RL benchmarks. There is little discussion on whether the approach generalizes to embodied agents or real-world robotics.

- Scalability concerns: Constructing and aligning latent graphs adds computational overhead. The paper briefly mentions efficiency optimizations, but these are not quantitatively assessed.

- Dependence on pretrained encoders: The reliance on pretrained visual backbones (e.g., DINO or MAE) raises the question of how performance scales when those are not available.

**Questions:**

- How sensitive is LGA to the number of latent factors? Does increasing or decreasing the number affect stability or learning speed?

- Can the graph alignment mechanism handle dynamic object counts, such as scenes where the number of entities changes over time?

- How does LGA behave in sparse reward settings where the supervision signal is weak or delayed?

- Have you tested how well LGA transfers between visually distinct environments that share similar causal structure?

- Is the computational cost of graph construction and alignment significant compared to baseline RL methods?

---

> ### Author Response · Authors · 2025-11-12
> **Potential review mixup?**
>
> Dear Reviewer ANPg,
>
> We suspect this review may have been posted here in error because it doesn't seem to be talking about our paper. There are references to methods, baselines and algorithms that are not discussed in our paper (e.g., LGA, DINO, MAE, DrQ, DMControl, Atari).

---

> > ### Comment · Reviewer_ANPg · 2025-11-12
> > **Sorry for the mixup**
> >
> > Thanks for catching it. I have reposted the review.

---

> ### Author Response · Authors · 2025-11-21
> **Thanks for your comments and recommendations**
>
> Thank you very much for your encouraging comments and thoughtful critique. We really appreciate it. Below we address your concerns.
>
> **Novelty**: We recognize that our work builds on prior ideas, and we cite those works as directly inspiring our approach. Our main contribution is a new algorithm that is derived from first principles. In contrast to ICF (Thomas et al, 2018), our method does not assume determinism and does not rely on reconstruction objectives. Moreover, our method was evaluated on environments with more factors, context dependent action effects and achieved perfect alignment in the 2D grid domain used in ICF (they only achieve identification up to a linear transformation with not theoretical guarantee).
>
> **On the simplicity of the evaluation domains:** We choose environments where (i) the ground-truth generative factors are fully accessible, (ii) actions influence different factors in a context-dependent and sparse way, and (iii) some factors are not independently controllable. These properties are essential for rigorously assessing whether ACF recovers the correct controllable factors. Although the environments we study are simple, they still present meaningful challenges for factorization and push the limits of the assumptions of ACF. For instance, in Taxi the passenger can only move with the taxi, therefore, the passenger is not independently controllable by the agent.
> Moreover, in Minigrid domains, the agent can move (relative to its orientation) forward, backward, left, right; this causes problems for many disentanglement algorithms because actions affect different variables depending on context. That is, there is no one-to-one correspondence between intervened variables and action labels.

---

> ### Author Response · Authors · 2025-11-21
> **Response II**
>
> **RL experiments and ACF consequences for RL**: Our goal in this paper is not to demonstrate downstream RL gains, but to evaluate ACF as a mechanism for discovering independently controllable factors from pixels. Prior RL work [1,3,5,6,7] already shows that when a factored representation is available, exploiting it leads to substantial improvements, but these works assume the representation is given. Our paper focuses on the foundational question of what that representation should be and whether it can be recovered reliably from observations. Once identified, these factors can be used by any factored or model-based RL method, but downstream RL evaluation is orthogonal to validating the representation itself. We will make this scope and motivation clearer in the revision.
>
> **Strong assumptions**: That’s a great question. Actions are of course not always sparse or independently controllable, but such assumptions are often realistic abstractions; agents typically cannot affect every variable at once nor do they want to. Action sparsity reflects the kinds of local, object-level actions we want agents to have. And if two variables cannot be independently controlled, it is unclear whether they should be treated as separate factors. Our aim is to introduce minimal domain-specific assumptions or human semantics. We agree controllable variables are not the only important aspects to disentangle, and future work will explore how other signals (e.g., reward) can support richer factorization.
>
> **Lack of stronger baselines**: To the best of our knowledge, we selected the methods that compete in performance and assumptions with ACF. Other methods for disentanglement rely on stronger assumptions (e.g., access to intervention masks [9], or one-to-one correspondence between action and intervened variable [8], access to known dynamics [10]). Moreover, within RL attempts are limited: they include beta-VAEs[1] (known to reliably identify variables [4]) and limited block factorization that separates variables from controllable-uncontrollable axes [2].
>
> **Multiple actions affecting the same variable**: We already explore this in Minigrid, where movement actions affect different dimensions depending on orientation. This is part of what makes FourRooms and similar domains more challenging than a simple 2D grid with north/south/east/west actions.
>
> **Scaling to more complex domains:** These more complex domains (e.g. Atari) we left for future work. However, the main complexity, in Atari, for example, is not visual complexity but because of the required long-term, temporally extended control—for instance, controlling the ball in Pong requires multi-step, complex behavior. Capturing such long-term effects is an exciting direction for future work.
>
> **Cost of contrastive learning:** Contrastive learning does not require more parameters than VAE-style methods, and while it is more computationally expensive, it remains tractable and offers more expressive modeling of transitions.
>
> **Weaker/Noisier interventions:**  Actions here can be seen as soft and stochastic interventions. A major strength of ACF is that we do not assume determinism, known interventions, or known dynamics, which gives the method high expressivity.
>
> **References**
> [1] Higgins, I., Pal, A., Rusu, A., Matthey, L., Burgess, C., Pritzel, A., Botvinick, M., Blundell, C., and
> Lerchner, A. Darla: Improving zero-shot transfer in reinforcement learning.
>
> [2] Wang, T., Du, S. S., Torralba, A., Isola, P., Zhang, A., and Tian, Y. Denoised mdps: Learning world
> models better than the world itself.
>
> [3] Wang, Z., Xiao, X., Xu, Z., Zhu, Y., and Stone, P. Causal dynamics learning for task-independent
> state abstraction
>
> [4]Locatello, F., Bauer, S., Lucic, M., Raetsch, G., Gelly, S., Schölkopf, B., and Bachem, O. Challenging
> common assumptions in the unsupervised learning of disentangled representations. In international
> conference on machine learning
>
> [5]Wang, Z., Hu, J., Stone, P., and Martín-Martín, R. Elden: Exploration via local dependencies.
> Advances in Neural Information Processing Systems
>
> [6]Wen Sun, Nan Jiang, Akshay Krishnamurthy, Alekh Agarwal, John Langford. Model-based RL in Contextual Decision Processes: PAC bounds and Exponential Improvements over Model-free Approaches
>
> [7]Strehl, A. L., Diuk, C., and Littman, M. L. Efficient structure learning in factored-state mdps.
>
> [8] Lippe, P., Magliacane, S., Löwe, S., Asano, Y. M., Cohen, T., and Gavves, E. Biscuit: Causal
> representation learning from binary interactions.
>
> [9] Lippe, P., Magliacane, S., Löwe, S., Asano, Y. M., Cohen, T., and Gavves, E. Causal representation
> learning for instantaneous and temporal effects in interactive systems.
>
> [10]Varici, B., Acartürk, E., Shanmugam, K., and Tajer, A. General identifiability and achievability for
> causal representation learning.

---

> ### Author Response · Authors · 2025-11-21
>
> We have revised our paper to address your concerns, and we hope these updates will support a reassessment of your score. We look forward to your response, and we are happy to answer any further questions you may have. If there are any remaining concerns that might hinder your reassessment, please let us know and we will do our best to address them.
> Thank you!

---

### Meta-Review · Area_Chair_6f4w · 2026-01-07

**Summary:**

Reviewers appreciated the clear motivation and theoretical contribution but were unsatisfied by the lack of downstream RL evaluation and the strong assumptions (bijective observations, no-op action). While the authors clarified the scope is identifiability, the paper's RL-centric framing led to expectations of practical utility demonstrations that weren't met.

**Reviewer Concerns:**

Addressed: The rebuttal clarified the paper's focus on identifiability, justified baseline selection, and explained that downstream RL gains are well-established in prior work, making their evaluation orthogonal.
Still Outstanding: The strong assumptions remain a practical limitation. The lack of RL experiments is still a missed opportunity to strengthen the paper's impact, even if the core claim is identifiability.

**Reviewer Scores:**

h8z1: Initially 4. The thoughtful rebuttal addressed many points, likely moving them to a 5 or 6.

2zgD: Initially 4. Remained concerned about RL evaluation; score likely unchanged at 4.

w7hc: Initially 6. Concerned about missing RL experiments; might lower to 5 due to the unresolved scope mismatch.

ANPg was reviewing a different paper, and their review was disregarded.

---

### Decision · Program_Chairs · 2026-01-26

Reject